# Neural representation of nouns and verbs in congenitally blind and sighted individuals

Marta Urbaniak ⓘ [1,2], Małgorzata Paczyńska[3], Alfonso Caramazza ⓘ [4,5,6] & Łukasz Bola ⓘ [1] ✉

In blind individuals, language processing activates not only classic language networks, but also the "visual" cortex. What is represented in visual areas when blind individuals process language? Here, we show that area V5/MT in blind individuals, but not other visual areas, responds differently to spoken nouns and verbs. We further show that this effect is present for concrete nouns and verbs, but not abstract or pseudo nouns and verbs. This suggests that area V5/ MT in blind individuals represents physical properties of noun and verb referents, salient in the concrete word category, but not conceptual or grammatical distinctions, present across categories. We propose that this motion-sensitive area captures systematically different motion connotations of objects (nouns) and actions (verbs). Overall, our findings suggest that responses to language in the blind visual cortex can be deconstructed to representing physical properties of words' referents, which are projected onto typical functional organization of this region.

Humans acquire language relatively quickly and effortlessly. This observation has generally been taken to suggest that the human brain has strong, innate adaptations to process language. One of these adaptations might be the evolution of the brain language network - a set of brain areas that have innate capability to process complex linguistic information. Neuropsychological and neuroimaging work supports the existence of such a network by showing that the neural basis of language processing is relatively robust to changes in individual experience. For example, similar brain regions are activated by words and sentences across a variety of spoken languages and cultures[1], and even across spoken and sign languages[2]. In all these cases, language processing primarily involves densely connected areas clustered around the left sylvian fissure, with the inferior frontal gyrus, the superior temporal lobe, and the anterior temporal lobe as key processing hubs[3]. Lesions or neurodegeneration in this network can cause deficits in language comprehension and production[3,4].

However, the view that language can be processed only by specialized brain areas has been challenged by the observation that, in blind individuals, linguistic stimuli activate not only the canonical language network, but also the "visual" cortex[5–10]. The magnitude of these activations increases with increasing complexity of linguistic stimuli[6,7,9], and is higher for semantic than phonological tasks[10]. Furthermore, a transient disruption of activity in early visual areas interferes with certain operations over linguistic stimuli in blind individuals, such as Braille reading[11,12] or word generation[13]. Finally, the functional connectivity between several visual regions and the classic language region, the left inferior frontal gyrus, is increased in blind individuals, compared to the sighted[7,14,15].

The theoretical implications of these findings are still debated, particularly because it is not clear what properties of linguistic stimuli are captured by the blind visual cortex. Previous studies have convincingly shown that, in the blind, this region is activated by a variety of linguistic stimuli and tasks. However, language can be represented at various levels, from relatively concrete representation of word and sentence referents (e.g., the word "an apple" refers to an object that is small, round, etc.) to abstract representations of predominantly linguistic properties (e.g., the word "apple" is a high-frequency noun). Processing of language at these different levels can produce

[1]Institute of Psychology, Polish Academy of Sciences, Warsaw, Poland. [2]Graduate School for Social Research, Polish Academy of Sciences, Warsaw, Poland. [3]Department of Psychology, SWPS University of Social Sciences and Humanities, Warsaw, Poland. [4]Department of Psychology, Harvard University, Cambridge, MA, USA. [5]Center for Mind/Brain Sciences (CIMeC), University of Trento, Trento, Italy. [6]Faculty of Psychology and Educational Sciences, University of Coimbra, Coimbra, Portugal. ✉e-mail: bolalukasz@gmail.com

deceptively similar patterns of activations: A word can induce stronger activity in a given brain area than a non-word either because it names an object with certain physical properties (compare "a ball" with "an allb") or because it is identified as a valid part of our lexicon; A sentence can induce stronger activity than a word either because it conveys more information about our physical environment (compare "a ball" with "Mary threw a ball to the dog") or because of its syntactic structure. It is still unclear at what level linguistic stimuli can be represented in the blind visual cortex, and to what extent this representation is truly different from those computed in the visual areas of the sighted.

In this study, we addressed this issue by investigating neural representations of nouns and verbs in congenitally blind and sighted individuals. The distinction between these two word classes is rich not only in abstract grammatical consequences, but also in relatively concrete semantic implications, with many nouns naming objects and many verbs naming actions. At the brain level, both nouns and verbs seem to be represented throughout the canonical language network in sighted individuals[16]. However, verbs elicit more activity in the left frontal and posterior temporal regions, whereas nouns induce stronger responses in the left inferior temporal regions[16–18]. Moreover, selective impairments in production of either nouns or verbs has been demonstrated by both neuropsychological[19] and neurostimulation[20,21] studies. This shows that, in the sighted brain, the processing of these two word classes is supported by at least partly different neural representations.

We enrolled congenitally blind and sighted participants in a functional magnetic resonance imaging (fMRI) experiment, in which they made morphological transformations (singular-to-plural number transformations) to spoken concrete nouns and verbs, abstract nouns and verbs, and morphologically-marked pseudo nouns and verbs. The words were chosen based on behavioral ratings by blind and sighted participants, and task difficulty was equated across nouns and verbs (see Methods and Supplementary Tables 1-4). The study used a block design (always presenting 6 words from the same category in a row) and different words were presented in each fMRI run. We used multi-voxel pattern classification to reveal brain areas representing the noun/verb distinction across and within the three semantic categories (concrete, abstract, pseudo) in both participant groups.

This design allowed us to disentangle grammatical and semantic representations in the visual cortex of blind individuals. Specifically, if the blind visual cortex represents differences between nouns and verbs at the grammatical level, we should observe comparable classification of activity patterns for nouns and verbs from all semantic categories in this region - in all categories, words were readily recognizable as nouns or verbs. Conversely, modulation of results by semantic category would suggest that the blind visual cortex represents differences between nouns and verbs at the semantic as opposed to the grammatical (morphosyntactic) level. To investigate these two possibilities, we controlled for phonological effects in all key analyses (see Methods).

We performed this study with two possibilities in mind. One possibility is that linguistic effects in the blind visual cortex are driven by, or are an extension of, typical visuospatial computations and processing hierarchies in this region. That would imply that the blind visual cortex represents linguistic stimuli at a relatively concrete level – one can assume, for example, that this region is sensitive to physical properties of objects and actions named by words and sentences. Based on this account, one might expect the activation patterns in the blind visual cortex to primarily reflect differences between concrete words, since a defining characteristic of these words is their having physical and spatial referents. Furthermore, if language effects in the blind visual cortex emerge within the typical functional organization of this region, then, in our study, these effects should be present in visual areas that are typically sensitive to physical properties that are differentially captured across nouns and verbs. A strong candidate for such a

region is area V5/MT[22,23], which is sensitive to visual[23] and auditory[24,25] motion. The auditory sensitivity of this area is elevated in blind individuals[26–28], and can potentially be used to represent motion connotations of nouns and verbs - a physical property that is differentially captured across these two word classes, especially in the concrete word category (see Supplementary Tables 3 and 4).

An alternative possibility is that responses to linguistic stimuli in the blind visual cortex are driven by a more fundamental form of neural plasticity. One can suppose, for example, that changes in relative strength of subcortical feedforward[29] and cortical backward projections[30] can lead to the development of new functional hierarchies and computational biases in this region. This can take the form of a "reverse hierarchy", in which the low-level visual areas assume more abstract cognitive functions than the high-level visual areas, which are richly multimodal and therefore "anchored" in their typical function even in blindness[31]. Such an account opens a possibility that, at least in certain visual areas in the blind, the distinction between nouns and verbs is represented at a more abstract, grammatical level. As was described above, in that case, the activation patterns in these areas should capture differences between nouns and verbs from all semantic categories (concrete, abstract, and pseudo) used in the study. Based on the "reverse hierarchy" hypothesis, one might expect to find such abstract representation in the low-level visual regions in blind individuals.

Our work supports the former hypothesis. We report different patterns of responses for nouns and verbs in area V5/MT in the blind participants, but not in other visual areas in this group. We further show that the effect in area V5/MT is present for concrete nouns and verbs, in the absence of significant results for abstract and pseudo nouns and verbs. This suggests that area V5/MT in blind individuals represents physical properties of noun and verb referents, salient in the concrete word category, but not conceptual or grammatical distinctions, present across the categories. We propose that this motion-sensitive area retrieves systematically different motion connotations of objects (nouns) and actions (verbs) from semantic representations computed in higher-level brain regions. Overall, our study suggests that responses to language in the blind visual cortex can be deconstructed to representing physical properties of words' referents, which are projected onto typical functional organization of this region.

## Results

### Multi-voxel pattern classification analysis

We first performed the classification of activity patterns for nouns and verbs in the visual areas when all semantic categories (concrete, abstract, pseudo) were included in one, omnibus analysis (Fig. 1; the masks of visual areas are presented in Supplementary Fig. 1). In the blind participants, this analysis produced a significant effect in area V5/MT (permutation analysis, mean classification accuracy = 54.3%, $p = 0.008$), but not in other visual areas studied (permutation analysis, all $p$ values > 0.14). In the sighted participants, no significant results were observed in any of the visual areas (permutation analysis, all $p$ values > 0.25). A direct between-group comparison confirmed that the classification of activity patterns for nouns and verbs in area V5/MT was more accurate in the blind group (two-tailed, two-sample $t$-test, mean difference = 5%, 95% CI [1.6%, 8.5%], t(38) = 2.94, $p = 0.006$, Cohen's d = 0.93).

We then asked whether the significant effect in area V5/MT in the blind participants could be driven by a specific word category. To address this question, we performed classifications of activity patterns for nouns and verbs in this area for each of the three word categories separately (Fig. 2). Indeed, in the blind participants, this analysis produced a significant effect for concrete nouns and verbs (permutation analysis, mean classification accuracy = 55.9%, $p = 0.024$), but not for abstract or pseudo nouns and verbs (permutation analysis, both $p$ values > 0.25). The classification accuracy for concrete nouns and

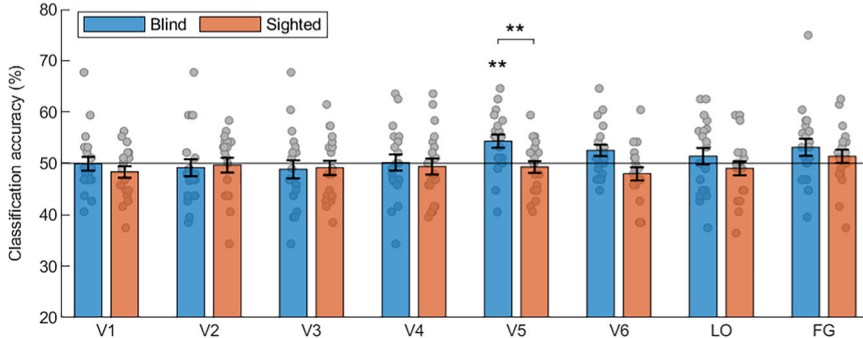

**Fig. 1 | The above-chance classification of activity patterns for nouns and verbs in area V5/MT in congenitally blind individuals.** Results of support vector machine classification of activity patterns for noun blocks and verb blocks in the visual areas in congenitally blind ($n = 20$) and sighted ($n = 20$) participants. LO the lateral occipital area, FG the fusiform gyrus. Statistical testing against classification chance level was performed separately in each participant group using the permutation procedure, in which the actual results were compared with the null distribution of 1000 classification values obtained with the labels of noun blocks and verb blocks randomly reassigned. The results were corrected for multiple comparisons across the visual areas, within each group (Bonferroni correction for 8 tests). The statistically significant results were found only in area V5/MT in blind individuals (mean classification accuracy = 54.3%, $**p = 0.008$). The between-group test was performed only in area V5/MT, in which significant results were observed in the blind group, using two-tailed, two-sample $t$-test. Since only one between-group comparison was performed, the correction for multiple comparisons was not necessary. This analysis confirmed that the classification of activity patterns for nouns and verbs in area V5/MT was more accurate in the blind group than in the sighted group (mean difference = 5%, 95% CI [1.6%, 8.5%], t(38) = 2.94, Cohen's d = 0.93, $**p = 0.006$). Error bars represent the standard error of the mean calculated across the results for individual participants in each group. The a priori chance classification level (accuracy = 50%) is marked with the black line. Source data are provided as a Source Data file.

verbs was significantly higher than the average classification accuracy calculated across the abstract and pseudo word categories (two-tailed, paired $t$-test, mean difference = 5.6%, 95% CI [0.9%, 10.3%], t(19) = 2.48, $p = 0.023$, Cohen's d = 0.55). In the sighted participants, the same analysis did not produce significant results in any word category (permutation analysis, all $p$ values > 0.25). However, a 2 (group) ×3 (word category) ANOVA indicated a significant main effect of word category (F(2,76) = 3.42, $p = 0.038$, $\eta_p^2 = 0.08$), with no main effect of group (F(1,38) = 1.27, $p = 0.267$, $\eta_p^2 = 0.03$) and no interaction between these two factors (F(2,76) = 0.33, $p = 0.717$, $\eta_p^2 = 0.01$). Thus, while the ANOVA confirmed the differences in the classification of activity patterns for nouns and verbs across the three word categories in area V5/MT, this analysis did not provide strong evidence that this effect is specific to the blind group.

As a control analysis, we investigated the accuracy of classification of activity patterns for nouns and verbs for each word category in all visual areas considered in the study (Supplementary Fig. 2). This analysis was meant to search for meaningful effects that were specific to only one word category and could have been potentially missed in the initial, omnibus analysis. However, apart from the already reported effect in area V5/MT in the blind participants, we only observed above-chance classification of activations for pseudo nouns and verbs in several visual areas in the blind participants (Supplementary Fig. 2). This result could perhaps be driven by some form of surprise response to these atypical stimuli. Critically, this effect was not accompanied by the successful classification of activations for abstract nouns and verbs in any of the visual areas (permutation analysis, all $p$ values > 0.25). Furthermore, besides the already reported effect in area V5/MT, the classification of activations for concrete nouns and verbs did not produce significant results in any other visual area (permutation analysis, all $p$ values > 0.25). This confirms the topographic specificity of the effect reported for concrete nouns and verbs in area V5/MT.

We investigated the robustness of our results in area V5/MT with several additional analyses. First, we asked whether the observed topographic specificity of the effect found in this area could be explained by the fact that its anatomical mask is smaller than the masks for other visual areas (see Supplementary Table 5). To investigate this issue, we iteratively drew the same number of voxels (193 voxels – the size of the V5 mask) from the mask of each visual region and reran our key comparisons in these subsets (Supplementary Figs. 3 and 4). We

replicated the results reported in the main analysis. This suggests that the reported results are not driven by the differences in the sizes of anatomical masks that were used.

Second, we wanted to rule out that the results observed in area V5/MT are driven by only a small subset of voxels within this area's mask – for example, only those voxels that border more anterior regions that represent actions at increasingly conceptual level[32]. To investigate this possibility, we iteratively drew subsets of voxels (40, 80, 120, and 160 voxels) from the V5/MT mask and performed the classification of activations for nouns and verbs from each semantic category in only these subsets (Supplementary Fig. 5). We found above-chance classification of activations for concrete nouns and verbs in blind individuals across all analysis levels (one-tailed, one-sample $t$-tests, analysis on 40 voxels: mean classification accuracy = 52.7%, 95% CI [50.1%, 55.3%], t(19) = 2.16, $p = 0.066$, Cohen's d = 0.48; analysis on 80 voxels: mean classification accuracy = 53.7%, 95% CI [50.4%, 57%], t(19) = 2.33, $p = 0.046$, Cohen's d = 0.52; analysis on 120 voxels: mean classification accuracy = 54.3%, 95% CI [50.4%, 58.2%], t(19) = 2.33, $p = 0.046$, Cohen's d = 0.52; analysis on 160 voxels: mean classification accuracy = 54.8%, 95% CI [50.5%, 59.1%], t(19) = 2.35, $p = 0.046$, Cohen's d = 0.53). Thus, this effect does not depend on several specific voxels within the V5/MT mask, and can be reliably detected across a broad spectrum of analysis parameters. In contrast, no significant effects, at any analysis level, were observed for abstract or pseudo nouns and verbs (Supplementary Fig. 5). As in the main analysis, no significant effects for nouns and verbs from any semantic category were observed in the sighted group (Supplementary Fig. 5).

Third, we reviewed the classification results obtained in area V5/MT for individual participants to ensure that our findings were not driven by outliers (Supplementary Fig. 6). In blind individuals, the above-chance classification of activations for concrete nouns and verbs in this area was observed in 14 out of 20 participants (70% of the participants). Across the results for three semantic categories in the blind group, 4 values were identified as potential outliers, defined as observations diverging from average classification accuracy in a given category by more than 2 standard deviations (1 value in the concrete category, 1 value in the abstract category, and 2 values in the pseudo category). Removing these values from the analysis further strengthened the group effects. Particularly, we still found above-chance classification of activations for concrete nouns and verbs (permutation

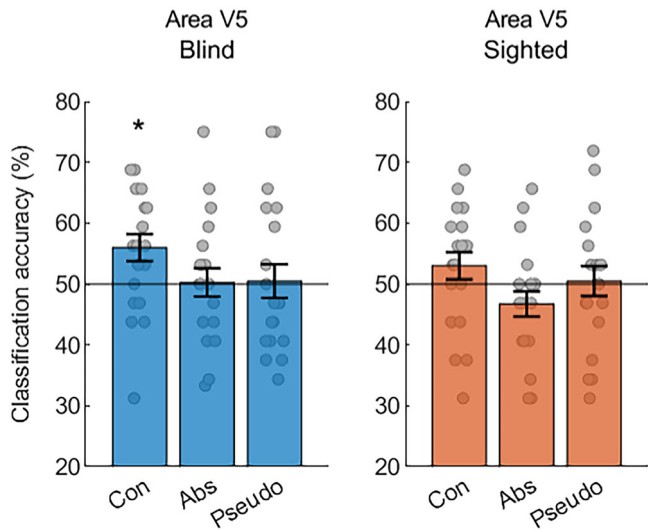

**Fig. 2 | The effect in area V5/MT in blind individuals is driven by successful classification of activity patterns for concrete nouns and verbs.** Results of support vector machine classification of activity patterns for noun blocks and verb blocks, performed separately for concrete, abstract, and pseudo word categories, in area V5/MT in congenitally blind (*n* = 20) and sighted (*n* = 20) participants. Statistical testing against classification chance level was performed separately in each participant group using the permutation procedure, in which the actual results were compared with the null distribution of 1000 classification values obtained with the labels of noun blocks and verb blocks randomly reassigned. The results were corrected for multiple comparisons across the word categories, within each group (Bonferroni correction for 3 tests). The statistically significant results were found only for the classification of concrete nouns and verbs in the blind group (mean classification accuracy = 55.9%, *p* = 0.024). Error bars represent the standard error of the mean calculated across the results for individual participants in each group. The a priori chance classification level (accuracy = 50%) is marked with black lines. Source data are provided as a Source Data file.

analysis, an increase in average classification accuracy from 55.9% to 57,2%, *p* = 0.003), but not for abstract and pseudo nouns and verbs (permutation analysis, both *p* values > 0.25). In the direct comparison across the categories we again found that the classification accuracy obtained for concrete nouns and verbs was significantly higher than the average classification accuracy calculated across the abstract and pseudo nouns and verbs (two-tailed, paired *t*-test, mean difference = 7.6%, 95% CI [2.8%, 12.3%], t(15) = 3.36, *p* = 0.004, Cohen's d = 0.84). In the sighted group, using the same data trimming procedure resulted in 2 values (1 value in the concrete category, 1 value in the abstract category) being removed from the data. Interestingly, after this procedure we found an uncorrected effect for concrete nouns and verbs also in area V5/MT in the sighted participants (permutation analysis, an increase in average classification accuracy from 53% to 54,1%, uncorrected *p* = 0.037), in the absence of effects for the abstract and pseudo nouns and verbs (permutation analysis, both uncorrected *p* > 0.25). The above-chance classification of activations for concrete nouns and verbs in this area was observed in 13 out of 20 sighted participants (13 out of 19 participants after data trimming, 68% of the participants). The direct comparison across semantic categories indicated a trend toward a higher classification accuracy for concrete nouns and verbs, compared to the average classification accuracy for abstract and pseudo nouns and verbs (two-tailed, paired *t*-test, mean difference = 6.3%, 95% CI [−0.2%, 12.7%], t(17) = 2.05, *p* = 0.056, Cohen's d = 0.48). Overall, the analysis of trimmed data suggests that, while statistically weaker, the results in area V5/MT in the sighted group might be qualitatively similar to those found in the blind group.

Next, we investigated what brain regions, beyond the visual cortex, capture the differences between nouns and verbs in the blind and

the sighted participants. To this aim, we performed an omnibus classification of activity patterns for nouns and verbs, with words from all three categories included, using a whole-brain searchlight approach (Fig. 3). In both participant groups, we observed significant effects primarily along the superior temporal sulci (Fig. 3a, b). In the blind participants, we additionally detected significant effects in the left inferior frontal gyrus (Brodmann areas 44 and 45), the left fusiform gyrus, and the left insula. A direct between-group comparison confirmed that the results in the opercular part of the left inferior frontal gyrus (Brodmann area 44) and in the insula were stronger in the blind group (Fig. 3c). There were no effects that were stronger in the sighted group.

We further asked whether the superior temporal cortex – the classic language region in which the omnibus analysis produced the strongest results in both groups – captures the differences between nouns and verbs for all three word categories included in the study. To answer this question, we performed an independent region of interest analysis in this area (Fig. 4). Indeed, we found above-chance classification of activations for nouns and verbs from all three word categories in both the blind and the sighted participants (permutation analysis, classification of concrete nouns and verbs in the blind group: *p* = 0.057; all other *p* values < 0.05; see also Supplementary Fig. 7), with no significant differences across the word categories in either blind group (two-tailed, paired *t*-tests, concrete vs. abstract: mean difference = −2.3%, 95% CI [−8.5%, 3.8%], t(19) = −0.8, *p* = 1, Cohen's d = −0.18; concrete vs. pseudo: mean difference = −0.7%, 95% CI [−8.3%, 6.9%], t(19) = −0.19, *p* = 1, Cohen's d = −0.04; abstract vs. pseudo: mean difference = 1.7%, 95% CI [−4.7%, 8.0%], t(19) = 0.55, *p* = 1, Cohen's d = 0.12) or sighted group (two-tailed, paired *t*-tests, concrete vs. abstract: mean difference = −0.3%, 95% CI [−6.6%, 5.9%], t(19) = −0.1, *p* = 1, Cohen's d = −0.02; concrete vs. pseudo: mean difference = −2.5%, 95% CI [−8.7%, 3.7%], t(19) = −0.84, *p* = 1, Cohen's d = −0.19; abstract vs. pseudo: mean difference = −2.2%, 95% CI [−9.7%, 5.3%], t(19) = −0.61, *p* = 1, Cohen's d = −0.14). A 2 (group) ×3 (word category) ANOVA did not produce any significant main effects or interactions (group: F(1,38) = 0.41, *p* = 0.529, $\eta_p^2$ = 0.01; word category: F(2,76) = 0.28, *p* = 0.755, $\eta_p^2$ = 0.01; group × word category: F(2,76) = 0.36, *p* = 0.698, $\eta_p^2$ = 0.01). This shows that the specific effect observed for concrete nouns and verbs in area V5/MT could not be explained by better classification of concrete words throughout the brain.

Additionally, we directly tested whether the result patterns in the superior temporal cortex and area V5/MT were different. Indeed, a 2 (brain area) × 2 (group) × 3 (word category) ANOVA produced a significant main effect of area (F(1,38) = 19.077, *p* < 0.001, $\eta_p^2$ = 0.33) and a trend-level interaction between the area and the word category factors (F(2,76) = 2.67, *p* = 0.076, $\eta_p^2$ = 0.07), with no other main effects or interactions being significant (group: F(1,38) = 0.13, *p* = 0.716, $\eta_p^2$ = 0; word category: F(2,76) = 1.05, *p* = 0.354, $\eta_p^2$ = 0.03; brain area × group: F(1,38) = 1.94, *p* = 0.172, $\eta_p^2$ = 0.05; group × word category: F(2,76) = 0.69, *p* = 0.505, $\eta_p^2$ = 0.02; brain area × group × word category: F(2,76) = 0.03, *p* = 0.973, $\eta_p^2$ = 0). The post hoc tests indicated that, compared to area V5/MT, the classification accuracy in the superior temporal cortex was significantly higher for the abstract (mean difference = 8.2%, 95% CI [4.2%, 12.1%], *p* < 0.001) and pseudo (mean difference = 6.5%, 95% CI [1.8%, 11.1%], *p* = 0.008) nouns and verbs, but not for concrete nouns and verbs (mean difference = 0.9%, 95% CI [−4.1%, 5.8%], *p* = 0.719).

Finally, the hypothesis that the blind visual cortex is sensitive to words because it represents the physical features of word referents implies that this region captures differences not only between concrete nouns and verbs, but also between concrete and abstract words. While our design was not optimized for this contrast, we tested this prediction and found a robust, above-chance classification of activity patterns for concrete and abstract words in areas V4 (permutation analysis, mean classification accuracy = 54.6%, *p* = 0.018) and V5

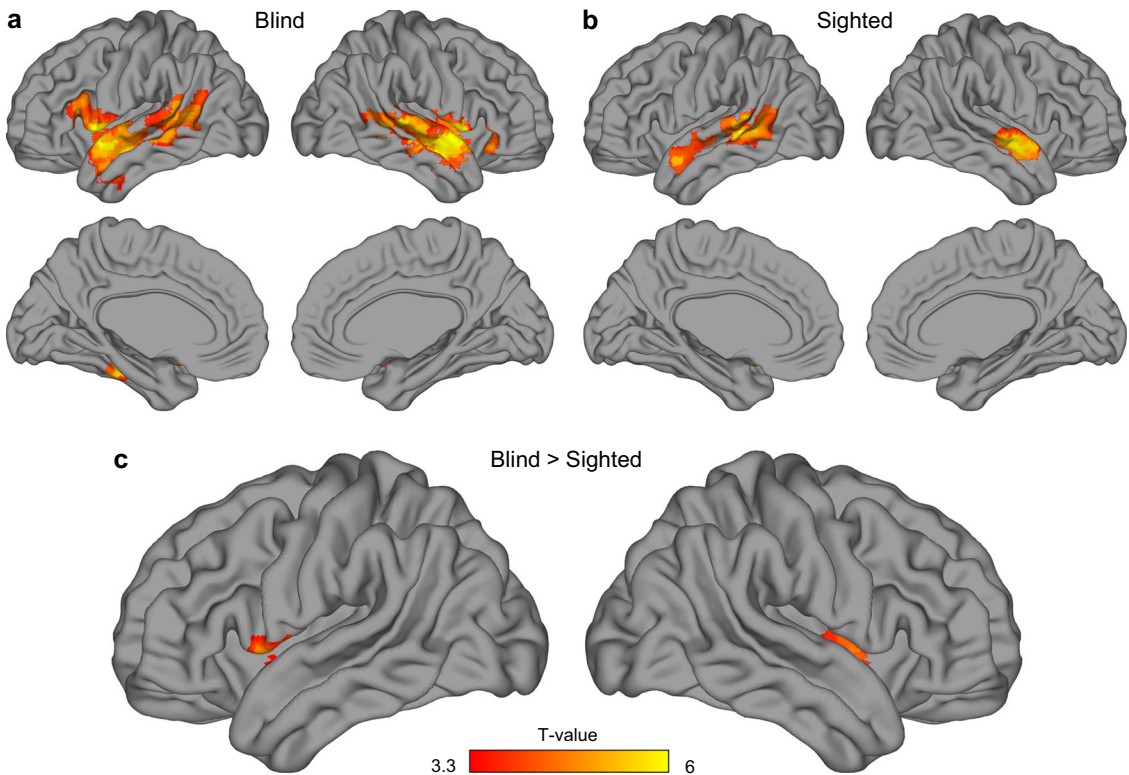

**Fig. 3 | A searchlight classification of activations for nouns and verbs.** Results of support vector machine classification of activity patterns for noun blocks and verb blocks, performed using a searchlight procedure **a** in congenitally blind participants (*n* = 20), **b** in sighted participants (*n* = 20), and **c** when the blind participants were compared to the sighted. Statistical testing against classification chance level (accuracy = 50%) in each participant group (**a**, **b**) was performed using one-tailed, one-sample *t*-tests. The between-group test (**c**) was performed using one-tailed, two-sample *t*-test. Statistical threshold for all these tests was set to *p* < 0.001, corrected for multiple comparisons using a family-wise error cluster correction. The results were visualized using BrainNet Viewer[70].

(permutation analysis, mean classification accuracy = 54.7%, *p* = 0.027) in the blind participants (Supplementary Fig. 8). Additionally, the effects in areas V1, V6, the lateral occipital area, and the fusiform gyrus in the blind group were significant at the uncorrected level (permutation analysis, all uncorrected *p* values < 0.05), but did not survive the correction for multiple comparisons. In contrast, the classification of activations for concrete and abstract words was not successful in the superior temporal cortex in this group, even at the uncorrected level (permutation analysis, uncorrected *p* = 0.249) (Supplementary Fig. 8). This further suggests that the effects observed in this classic language region might be driven by different neural representation than the effects found in the blind visual cortex.

## Univariate analysis

We also performed univariate analyses to verify whether the visual cortex in blind participants responded to spoken words more strongly than the visual cortex in sighted participants. The whole-brain analysis (Fig. 5) showed that, in both groups, listening to spoken words and pseudowords activated the classic language regions (Fig. 5a, b). In the blind participants, significant responses to these stimuli were also observed in the high-level visual areas. The direct between-group comparison showed that both early and high-level visual areas responded to spoken words more strongly in the blind group than in the sighted group (Fig. 5c). An additional region of interest analysis (Supplementary Fig. 9) showed that this effect was driven by both activations in the visual cortex in the blind participants and deactivations in the visual cortex in the sighted participants.

Furthermore, we investigated univariate activations elicited by specific word classes in area V5/MT (Fig. 6), in which multi-voxel activation patterns were different for concrete nouns and verbs. In the

blind participants, we observed activation of this area for all word classes, compared to rest periods (two-tailed, one-sample *t*-tests, concrete nouns: mean contrast estimate = 1.11, 95% CI [0.32, 1.91], t(19) = 2.93, *p* = 0.054, Cohen's d = 0.66; concrete verbs: mean contrast estimate = 1.26, 95% CI [0.41, 2.12], t(19) = 3.09, *p* = 0,036, Cohen's d = 0.69; abstract nouns: mean contrast estimate = 1.26, 95% CI [0.53, 1.99], t(19) = 3.61, *p* = 0.012, Cohen's d = 0.81; abstract verbs: mean contrast estimate = 1.25, 95% CI [0.36, 2.13], t(19) = 2.94, *p* = 0.048, Cohen's d = 0.66; pseudo nouns: mean contrast estimate = 2.25, 95% CI [1.41, 3.09], t(19) = 5.62, *p* < 0.001, Cohen's d = 1.26; pseudo verbs: mean contrast estimate = 2.31, 95% CI [1.44, 3.18], t(19) = 5.56, *p* < 0.001, Cohen's d = 1.24). In the sighted participants, in contrast, all word classes induced deactivation of this region, relative to rest periods (two-tailed, one-sample *t*-tests, concrete nouns: mean contrast estimate = −0.8, 95% CI [−1.31, −0.29], t(19) = 3.27, *p* = 0.024, Cohen's d = 0.73; concrete verbs: mean contrast estimate = −0.83, 95% CI [−1.46, −0.19], t(19) = 2.7, *p* = 0.084, Cohen's d = 0.61; abstract nouns: mean contrast estimate = −0.86, 95% CI [−1.45, −0.27], t(19) = 3.04, *p* = 0.042, Cohen's d = 0.68; abstract verbs: mean contrast estimate = −1.15, 95% CI [−1.76, −0.53], t(19) = 3.91, *p* = 0.006, Cohen's d = 0.88; pseudo nouns: mean contrast estimate = −1.5, 95% CI [−2.04, −0.95], t(19) = 5.78, *p* = <0.001, Cohen's d = 1.29; pseudo verbs: contrast estimate = −1.07, 95% CI [−1.68, −0.46], t(19) = 3.67, *p* = 0.012, Cohen's d = 0.82). A 2 (group) × 2 (grammatical class) × 3 (word category) ANOVA indicated a significant main effect of group (F(1,38) = 37.36, *p* < 0.001, $\eta_p^2$ = 0.5), a significant main effect of word category (F(2,76) = 4.16, *p* = 0.019, $\eta_p^2$ = 0.1), and an interaction between these two factors (F(2,76) = 18.21, *p* < 0.001, $\eta_p^2$ = 0.32). Interestingly, neither main effect of grammatical class nor interactions including this factor were significant (grammatical class: F(1,38) = 0.23, *p* = 0.632, $\eta_p^2$ = 0.01; group × grammatical class: F(1,38) = 0.18, *p* = 0.893, $\eta_p^2$ = 0;

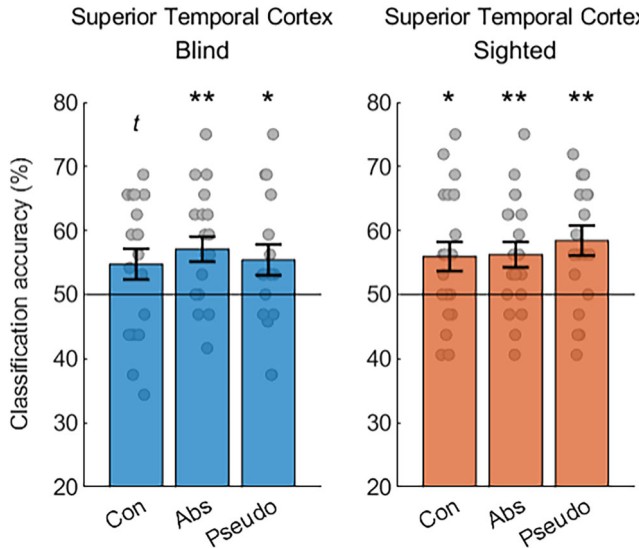

**Fig. 4 | The effects in the superior temporal cortex are driven by successful classification of activity patterns for nouns and verbs from all semantic categories.** Results of support vector machine classification of activity patterns for noun blocks and verb blocks, performed separately for concrete, abstract, and pseudo word categories, in the superior temporal cortex in congenitally blind ($n = 20$) and sighted ($n = 20$) participants. Statistical testing against classification chance level was performed separately in each participant group using the permutation procedure, in which the actual results were compared with the null distribution of 1000 classification values obtained with the labels of noun blocks and verb blocks randomly reassigned. The results were corrected for multiple comparisons across the word categories, within each group (Bonferroni correction for 3 tests). The statistically significant results were found in both the blind group (concrete nouns and verbs: $^t p = 0.057$; abstract nouns and verbs: $^{**}p = 0.006$; pseudo nouns and verbs: $^*p = 0.027$) and the sighted group (concrete nouns and verbs: $^*p = 0.018$; abstract nouns and verbs: $^{**}p = 0.009$; pseudo nouns and verbs: $^{**}p = 0.003$). Error bars represent the standard error of the mean calculated across the results for individual participants in each group. A priori chance classification level (accuracy = 50%) is marked with black lines. Source data are provided as a Source Data file.

grammatical class × word category: $F(2,76) = 1.5$, $p = 0.23$, $\eta_p^2 = 0.04$; group × grammatical class × word category: $F(2,76) = 1.17$, $p = 0.315$, $\eta_p^2 = 0.03$. The post-hoc tests showed that, in the blind participants, pseudowords induced stronger activation in area V5/MT than concrete words (mean difference = 1.09, 95% CI [0.56, 1.62], $p < 0.001$) and abstract words (mean difference = 1.03, 95% CI [0.53, 1.54], $p < 0.001$), in the absence of significant difference between concrete and abstract words (mean difference = −0.06, 95% CI [−0.5, 0.38], $p = 1$). In contrast, no differences across word categories were observed in the sighted participants (concrete vs. abstract: mean difference = 0.19, 95% CI [−0.25, 0.63], $p = 0.852$; concrete vs. pseudo: mean difference = 0.47, 95% CI [−0.06, 1], $p = 0.096$; abstract vs. pseudo: mean difference = 0.28, 95% CI [−0.23, 0.79], $p = 0.517$). There were no differences between nouns and verbs in any word category in both the blind participants (concrete nouns vs. concrete verbs: mean difference = −0.15, 95% CI [−0.64, 0.34], $p = 0.541$; abstract nouns vs. abstract verbs: mean difference = 0.01, 95% CI [−0.471, 0.495], $p = 0.96$; pseudo nouns vs. pseudo verbs: mean difference = −0.06, 95% CI [−0.53, 0.41], $p = 0.807$) and the sighted participants (concrete nouns vs. concrete verbs: mean difference = 0.03, 95% CI [−0.46, 0.52], $p = 0.913$; abstract nouns vs. abstract verbs: mean difference = 0.29, 95% CI [−0.19, 0.77], $p = 0.229$; pseudo nouns vs. pseudo verbs: mean difference = −0.43, 95% CI [−0.9, 0.04], $p = 0.074$). These results suggest that global, univariate responses might capture different neural processes in area V5/MT of blind individuals than the multi-voxel activation patterns.

## Exploratory analyses

We tested whether the effects observed in the classification analysis in area V5/MT were lateralized to the language-dominant hemisphere. Given that the lateralization of the language network is more variable in blind individuals than in sighted individuals[6,9,33,34], we empirically determined which hemisphere is language-dominant in each participant by comparing the magnitudes of activations for words and pseudowords in classic language regions (superior temporal and inferior frontal cortices) and in their analogs in the right hemisphere (see Methods). For each participant, we then ran separate classification analysis in area V5/MT in the language-dominant and the language-nondominant hemisphere. However, the group effects obtained in these analyses (Supplementary Fig. 10) were weaker than those produced by the bilateral analysis. In the language-dominant hemisphere, we observed uncorrected effects in the omnibus classification of activity patterns for all nouns and all verbs in both blind individuals (permutation analysis, mean classification accuracy = 52.3%, uncorrected $p = 0.038$) and sighted individuals (permutation analysis, mean classification accuracy = 52%, uncorrected $p = 0.049$). However, these results did not survive the correction for multiple comparisons (permutation analysis, both $p$ values > 0.15). Furthermore, no significant results, in either group and hemisphere, were detected in the more detailed analysis, in which we classified activity patterns for nouns and verbs from each semantic category separately (permutation analysis, all $p$ values > 0.1). This suggests that the robust results reported in area V5/MT in blind individuals in the main analysis were driven by activity patterns in both hemispheres. One might speculate that the analysis in specific hemispheres lacked power to detect robust effects, akin to those detected in the bilateral analysis.

We also further investigated whether differences between nouns and verbs were represented in high-level ventral and ventrolateral visual areas. To this aim, we ran our analyses in the lateral occipito-temporal cortex (LOTC) and the ventral occipitotemporal cortex (VOTC), as defined by previous studies that reported categorical effects in these regions in blind individuals[35,36] (Supplementary Fig. 11). In the omnibus classification of activity patterns for all nouns and verbs, we found a significant effect in the VOTC in the blind group (permutation analysis, mean classification accuracy = 53.7%, $p = 0.024$). However, the more detailed analysis suggested that this effect is driven primarily by successful classification of pseudo nouns and verbs (permutation analysis, mean classification accuracy = 55.9%, $p = 0.02$), in the absence of significant results for concrete and abstract nouns and verbs (permutation analysis, both $p > 0.25$). We did not find robust effects in the LOTC in either group (permutation analysis, all $p$ values > 0.1) or in the VOTC in the sighted group (permutation analysis, all $p$ values > 0.25). Overall, the differences between concrete or abstract nouns and verbs do not seem to be robustly represented in ventral or ventrolateral visual areas.

## Discussion

In this study, we found that classification of activity patterns for nouns and verbs was significantly above chance level in area V5/MT in congenitally blind participants, but not in other visual areas in this group. We further showed that the effect in area V5/MT in the blind was driven by successful classification of activations for concrete nouns and verbs, in the absence of significant results for abstract and pseudo nouns and verbs. Beyond the visual cortex, we found successful classification of activity patterns for nouns and verbs primarily in the superior temporal cortex, in both blind and sighted participants, with additional effects in several other classic language regions in the blind group. In both groups, the activity patterns in the superior temporal cortex captured differences between nouns and verbs from all three word categories (concrete, abstract, pseudo) used in the study.

Words are highly multidimensional objects. A given brain area can represent differences between words because of how they sound

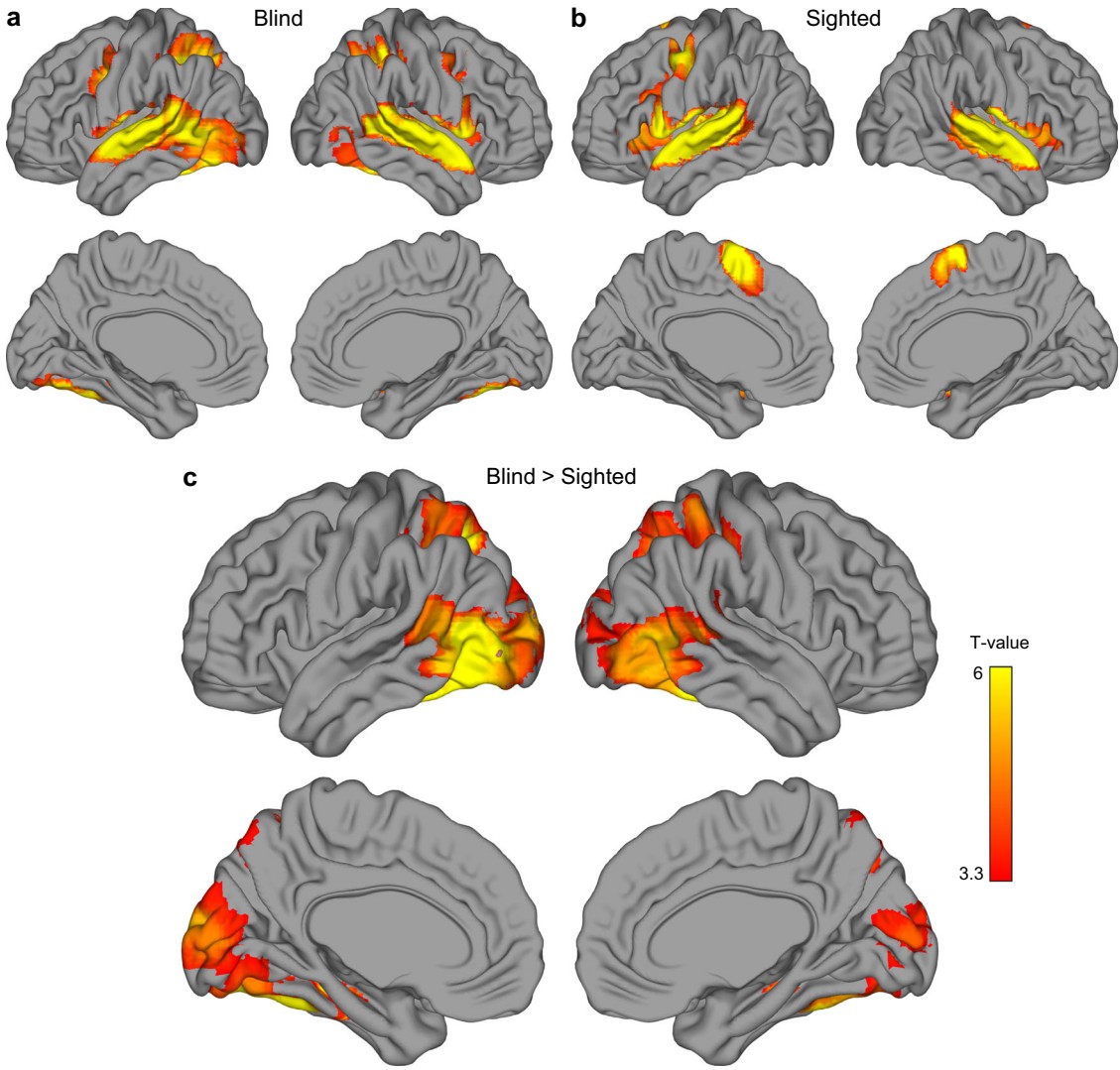

**Fig. 5 | Brain activations for spoken words in congenitally blind and sighted individuals.** Average responses to all spoken words and pseudowords, compared to activation during rest periods, **a** in congenitally blind participants (*n* = 20), **b** in sighted participants (*n* = 20), and **c** when the blind participants were compared to the sighted. Statistical testing against activation during rest in each participant group (**a**, **b**) was performed using one-tailed, one-sample *t*-tests. The between-group test (**c**) was performed using one-tailed, two-sample *t*-test. Statistical threshold for all these tests was set to *p* < 0.001, corrected for multiple comparisons using a family-wise error cluster correction. The results were visualized using BrainNet Viewer[70].

(phonological dimension), what they mean (semantic dimension), or what role they play in a sentence (grammatical dimension). The aim of our study was to investigate which word properties are represented in the visual cortex of blind individuals, and could drive activation for linguistic stimuli in this region[5–10]. We focused on a fundamental linguistic distinction, that between nouns and verbs, and investigated whether semantic or grammatical aspects of this distinction are represented in the blind visual cortex. We found above-chance classification of activity patterns for nouns and verbs in visual area V5/MT in the blind participants, but not in other visual areas in this group. We further showed that the effect in area V5/MT in the blind was primarily driven by successful classification of activations for concrete nouns and verbs, in the absence of significant results for abstract and pseudo nouns and verbs. Different classification results for nouns and verbs from different semantic categories cannot be explained by differences in auditory representations, as the phonological properties of words were not systematically different across the semantic categories. Similarly, this pattern of results cannot be driven by grammatical representations – in all semantic categories, words were readily recognizable as nouns or verbs. Thus, these results suggest that area

V5/MT in blind individuals represents differences between nouns and verbs because of their differing semantic properties, that is, through representations of objects and actions named by words.

Our study further shows that area V5/MT in blind individuals captures properties that are saliently and systematically different for concrete noun and verb referents, but not necessarily for abstract noun and verb referents. We suggest that the most plausible candidate for such a property is the representation of motion connotations, which is likely differently activated by concrete nouns, generally naming stationary objects, and concrete verbs, generally naming dynamic actions. Area V5/MT is sensitive to visual[23] and auditory[24,25] motion, with auditory sensitivity being preserved and elevated in blind individuals[26–28]. Our study suggests that this area can also retrieve motion connotations of objects and actions from semantic representations. In other words, our findings indicate that this area can retrieve the same physical property from memory-driven semantic representations and stimulus-driven perceptual (visual, auditory, etc.) representations.

This conclusion is in line with previous reports that, in both blind and sighted individuals, high-level ventral visual regions respond

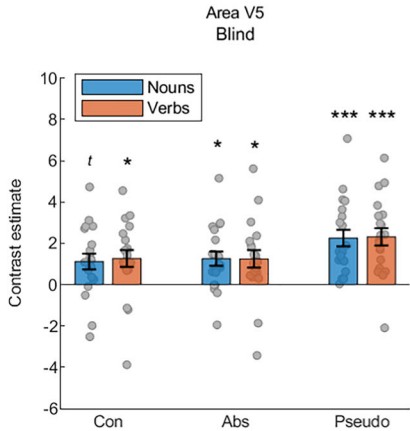
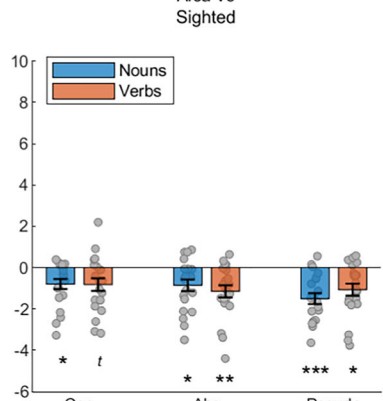

**Fig. 6 | Brain responses to specific word categories in area V5/MT in congenitally blind (*n* = 20) and sighted (*n* = 20) individuals.** The responses are presented relative to activations during rest periods. Statistical testing against rest-level activations was performed separately in each participant group using two-tailed, one-sample *t*-tests. The results were corrected for multiple comparisons across the word categories, within each group (Bonferroni correction for 6 tests). In the blind group, the analysis indicated statistically significant increases of activation magnitude, relative to rest periods (concrete nouns: *ᵗp* = 0.054; concrete verbs:

*p* = 0.036; abstract nouns: *p* = 0.012; abstract verbs: *p* = 0.048; pseudo nouns: ***p* = 0.0001; pseudo verbs: ***p* = 0.0001). In contrast, in the sighted group, the analysis revealed statistically significant decreases of activation, relative to rest periods (concrete nouns: *p* = 0.024; concrete verbs: *ᵗp* = 0.084; abstract nouns: *p* = 0.042; abstract verbs: ***p* = 0.006; pseudo nouns: ****p* = 0.0001; pseudo verbs: *p* = 0.012). Error bars represent the standard error of the mean calculated across the results for individual participants in each group. Source data are provided as a Source Data file.

differently to words referring to objects of different shape and size[37]. Similar effects were shown in these regions during the presentation of object pictures to sighted participants[38]. Importantly, representation of fruit color, following the spoken presentation of fruit names, was documented in the ventral visual cortex of sighted individuals, but not congenitally blind individuals[39]. This result suggests that visual areas only represent object physical features that are grounded in individual perceptual experience, also in blind individuals – abstract semantic knowledge that "apples are red, and similar in color to strawberries" is not represented in the visual cortex in this population. These findings concur with the results of our study. First, we show that also dorsal visual regions can use information conveyed by spoken words to perform their relatively typical computations, such as representation of motion and motion connotations. Second, we used abstract and pseudo words to test for more abstract, conceptual or grammatical representations in the blind visual cortex, as such representations could be potentially computed on top of simpler representations of physical properties. However, we did not find any clear sign of such abstract representations in any visual area tested.

Overall, our results suggest that, during language processing, the blind visual cortex represents the physical properties of word referents, more salient in the concrete word category, rather than more abstract linguistic features, present across the word categories. The topography of effects observed in our study – that is, finding the representation of differences between two word classes with systematically different motion connotations primarily in the motion-sensitive area V5/MT - indicates that these physical connotations conveyed by words are mapped onto the typical functional organization of the visual cortex, present also in the sighted brain.

Our findings suggest that, after the activation of semantic representations in higher-level brain regions, the physical properties of word referents, retrieved from these representations, are back projected to the visual system in a way that parallels feedforward visual processing. Thus, the "motion template" of objects and actions is back projected to the dorsal stream areas, such as area V5/MT, the "shape template" is back projected to the ventral stream areas, and so on. Such organization of the back projections from the semantic system might be most useful for forming visual predictions and, consequently, supporting visual perception[40]. This mechanism might be preserved and functional in blind individuals, even if its original function is not

relevant in this population. Moreover, the impact of such modulations on the visual cortex activity might be greater in blindness because of the lack of competition from feedforward visual inputs and weakening of inhibitory mechanisms in the visual areas in this population[41,42]. This view implies that our results were restricted to area V5/MT not because this area is especially sensitive to semantic properties of words, compared to other visual areas. Instead, stronger results in this area were observed simply because we classified activity patterns for nouns and verbs, that is, two word classes with differing motion connotations. Other contrasts should reveal representations of other physical properties of word referents in other visual areas in blind individuals. In line with this prediction, the broad contrast between all concrete words (with physical referents) and all abstract words (without physical referents) produced results in both dorsal and ventral visual areas in blind individuals, and even in the primary visual cortex (Supplementary Fig. 8). Furthermore, based on this hypothesis, qualitatively similar, even if weaker results should be expected in sighted individuals. In line with this prediction, our analysis revealed statistical trends in area V5/MT in the sighted participants (Supplementary Fig. 6), which are qualitatively similar to the results observed in this area in the blind participants.

The above-described view about activations induced in the blind visual cortex by language processing concurs with findings in the perceptual domain. Research in this domain shows that the processing of auditory and tactile stimuli by blind individuals activates specific visual areas that, in sighted individuals, process comparable stimuli in the visual modality[43–45]. Some of these auditory and tactile effects were also detected in the visual cortex of sighted individuals[24,25,46–48]. Based on these results, it has been suggested that many visual areas can retrieve comparable information from perceptual experiences in different sensory modalities. Here, we show that this "computational equivalency" principle may also organize back projections from higher-level semantic regions to the visual cortex.

Our findings contribute to a better understanding of principles of plasticity in the human brain. One way to think of language-driven activations in the blind visual cortex is that, in the absence of visual signals, this region develops new functional properties, which are radically different from those computed in the sighted visual cortex[49]. Our findings provide a different perspective and suggest that at least some effects observed for language in the blind visual cortex might be

explained by typical computational biases and preserved ability of this region to compute physical and spatial representations of the world. In this view, responses to linguistic stimuli in the blind visual cortex can be driven by strengthening and uncovering of functional interactions between higher-order and visual cortices that are present also in the sighted brain[43].

The univariate analyses reported here showed activations for spoken words in the visual cortex in blind participants, a result that is consistent with previous studies. Interestingly, the analysis in area V5/MT in this group suggests that the global magnitude of activation in this region, as measured by univariate methods, captures different processes than the multi-voxel activation patterns. The global activation of area V5/MT in the blind was the same for all word classes, but higher for pseudowords, which suggests that this measure captured increased processing demands related to surprising or atypical stimuli. Attention or cognitive load modulates the activation magnitude in the visual cortex even in sighted individuals[50,51]. The increase in sensitivity of the blind visual cortex might strengthen these effects, leading to the generic response that can cover semantically-driven processes, specific for a given word category. These more specific processes can nevertheless be detected with multivariate analysis. One possibility is that the difference in results obtained with these two analytical approaches is driven entirely by differences in statistical power. However, it is also possible that more complex effects are truly represented at the level of large-scale activity patterns, which cannot be detected even by the most powerful univariate (i.e., voxel-by-voxel) analysis. Notably, such disparity in results of these two analytical approaches, with the multivariate analysis showing results that were not detected in the univariate analysis, were reported in several recent studies of the blind visual cortex that did not use linguistic stimuli[52,53]. Thus, this may be a more general phenomenon, not specifically tied to linguistic processing.

Conversely, in the visual cortex of sighted individuals, spoken words induced robust deactivation, relative to the rest periods. This effect might be driven by a known mechanism of inhibition of the visual system activity during auditory perception[54,55]. This process might be less efficient in blind individuals due to generally weaker inhibition mechanisms in their visual areas, described above. One study demonstrated that certain regions within the early visual cortex of sighted individuals are activated by spoken words[56]. Here, we did not find such an effect. One explanation could be the differences in tasks used in the two studies. In the first experiment, Seydell-Greenwald and colleagues asked the participants to listen to sentences and decide whether they are semantically correct (e.g., "a big gray animal is an elephant"). In the two remaining studies, the authors asked the participants to listen to words and detect rare semantic oddballs (i.e., fruit names). In other words, in all experiments, the participants were encouraged to think about word meaning. In our study, we used a morphological transformation task, which focused the participants' attention on word grammatical properties. Perhaps attention directed to word meaning can strengthen the modulation of the visual cortex by semantic regions, and result in above-rest activations for words also in the sighted visual cortex.

Beyond the visual cortex, we observed the representation of differences between nouns and verbs only in the classic language regions, in both the blind and the sighted participants. In both groups, this representation was strongest in the middle and superior temporal cortices. This region is involved in perceptual analysis of auditory signals, which also includes parsing speech into phonemes, words, and expressions[57]. However, besides perceptual processing of speech, the superior temporal cortex also computes semantic and grammatical representations, which are largely independent of auditory properties of speech themselves[3]. In our study, auditory processing could not affect the classification of activity patterns for nouns and verbs in the searchlight analysis, as phonology of words was systematically altered

across the odd and even runs, and the classification was performed in the odd/even cross-validation scheme (see Methods). Thus, the results observed in the searchlight analysis in the superior temporal cortex must be driven by linguistic representations that are independent of phonological processing. Apart from the superior temporal cortex, there were several, additional effects in other classic language regions in the blind participants. One possible explanation is that blind individuals are more attentive to speech[58], which can lead to more robust signal in certain language areas. Despite these between-group differences, our results suggest that the overall topography of the classic language network is robust to changes in visual experience.

Two limitations of our work should be acknowledged. First, the searchlight analysis did not confirm the results obtained in area V5/MT in the analysis in visual regions of interest. This difference in results across these two methods is likely driven by the fact that the searchlight analysis included V5/MT voxels from only one hemisphere at a time. Our analyses show that only including activity patterns from both hemispheres results in robust effects in this area. Second, we did not confirm that area V5/MT represents motion connotations of word referents with representational similarity analysis. Our study was optimized for multi-voxel pattern classification and used a block design and different words in each experimental run. While these methodological choices served an important purpose in our study, they also precluded the representational similarity analysis.

In conclusion, our findings suggest that the blind visual cortex represents the physical properties of word referents rather than more abstract, conceptual or grammatical distinctions. This shows that at least some effects observed for language in the blind visual cortex might be explained by preserved ability of this region to compute physical and spatial representations of the world. The topography of effects observed in our study suggests that, in blind individuals, physical connotations conveyed by spoken words are mapped onto the typical functional organization of the visual cortex.

## Methods
### Participants
Twenty congenitally blind subjects (9 males, 11 females, mean age = 35.65 y, SD = 7.81 y, average length of education = 14.8 y, SD = 2.35 y) and 20 sighted subjects (6 males, 14 females, mean age = 35 y, SD = 8.58 y, average length of education = 15.4 y, SD = 2.04 y) participated in the study. The sex of participants was determined based on self-report. The sex-based analyses were not performed because of relatively low sample size and lack of a priori hypotheses concerning the impact of this dimension on the results. All except two participants were right-handed, and the remaining two participants (one blind, one sighted) were left-handed. The blind and the sighted groups were matched for age (two-tailed Mann–Whitney test, U = 195, p = 0.904), sex (two-tailed chi-square test, $X^2$ = 0.96, p = 0.327), handedness (two-tailed chi-square test, $X^2$ = 0, p = 1), and years of education (two-tailed Mann–Whitney test, U = 174, p = 0.495). In the blind group, blindness had a variety of causes, including retinopathy of prematurity, glaucoma, Leber's congenital amaurosis, optic nerve hypoplasia, or unknown causes. Most blind participants reported to have some light perception, but no object or contour vision. One blind participant reported to have some form of contour vision, which, however, was not precise enough to be functional. All subjects in both groups were native Polish speakers, had normal hearing, and had no history of neurological disorders. All subjects had no contraindications to the MRI, gave written informed consent and were paid for participation. The study was approved by the ethics committee of Institute of Psychology, Polish Academy of Sciences.

### Stimuli
In total, 144 stimuli were used: 24 concrete nouns, 24 concrete verbs, 24 abstract nouns, 24 abstract verbs, 24 pseudo nouns, and 24 pseudo

verbs. All stimulus categories were matched on average number of syllables, and all words categories were additionally matched on average frequency of occurrence in Polish language (quantified as Zipf score[59]), as indicated by Subtlex-pl database[60] (Supplementary Table 1). The length and the frequency matching were performed not only for the stimulus forms heard by the participants, but also for the target forms that the participants were expected to produce in the fMRI (Kruskal–Wallis tests, number of syllables in the heard word forms: H(5) = 1.23, $p$ = 0.942; number of syllables in the target word forms: H(5) = 0.89, $p$ = 0.971; frequency of the heard word forms: H(3) = 2.95, $p$ = 0.399; frequency of the target word forms: H(3) = 0.362, $p$ = 0.948). All chosen words had to fulfill semantic criteria – broadly, all concrete words referred to objects or actions that are well specified in space (e.g., "a cup" or "to kick"), whereas all abstract words referred to concepts or conceptual actions without a clear spatial framework (e.g., "fairness" or "to think"; see the Supplementary Table 6 for a complete list of stimuli and their translations). All pseudowords were phonologically and grammatically valid but had no meaning in Polish. The pseudowords were created by mixing syllables taken from the actual words. The stimuli were audio recorded using speech synthesizer software. The recordings were judged as sounding natural and readily understandable by Polish native speakers during pilot studies.

## Behavioral experiments

The final stimuli were chosen from a larger initial dataset (240 items, 40 per category) based on the results of two pilot behavioral studies. In the first study, 15 sighted participants (8 males, 7 females, mean age = 27.6 y, SD = 7.94 y, average length of education = 14.67 y, SD = 3.5 y) were asked to transform the heard words and pseudowords, from singular to plural form, based on the verbal transformation cues, the same as were used in the fMRI experiment (see the "fMRI experiment" section below). Each item was repeated two times (480 trials in total) and the subjects were asked to produce an overt response, which was recorded using a microphone. Next, response times for each item (from the onset of word presentation to the onset of response) were calculated using Chronset[61]. The response times across stimulus categories were matched as well as possible. In the final stimulus list, response times were matched across all word categories (Kruskal–Wallis test, H(3) = 3.16, $p$ = 0.368) and across pseudo nouns and pseudo verbs (two-tailed Mann–Whitney test, U = 278, $p$ = 0.837) (see Supplementary Table 2 for average response times for all stimulus categories in the final stimulus list). As could be expected, the response times for pseudo nouns and pseudo verbs were higher than for any word category (two-tailed Mann–Whitney tests, pseudo nouns vs. concrete nouns: U = 69, $p$ < 0.001; pseudo nouns vs. concrete verbs: U = 88, $p$ < 0.001; pseudo nouns vs. abstract nouns: U = 38, $p$ < 0.001; pseudo nouns vs. abstract verbs: U = 40, $p$ < 0.001; pseudo verbs vs. concrete nouns: U = 174, $p$ = 0.019; pseudo verbs vs. concrete verbs: U = 140, $p$ = 0.002; pseudo verbs vs. abstract nouns: U = 145, $p$ = 0.003; pseudo verbs vs. abstract verbs: U = 136, $p$ = 0.002).

In the second study, 15 congenitally blind (5 males, 10 females, mean age = 36.27 y, SD = 7.89 y) and 46 sighted participants (23 males, 23 females, mean age = 25.46 y, SD = 7.69 y) were asked to rate each word from the initial list on three scales (from 1 to 7): concreteness, imaginability, and movement connotations. Then, the items with unexpected ratings (e.g., concrete words with relatively low concreteness scores) were excluded. In the final stimulus set, all word categories were rated as expected by both groups – that is, concreteness and imaginability scores were higher for the concrete words than for the abstract words (two-tailed Mann–Whitney tests, concreteness ratings in the blind group: U = 0, $p$ < 0.001; imaginability ratings in the blind group: U = 0, $p$ < 0.001; concreteness ratings in the sighted group: U = 0, <0.001; imaginability ratings in the sighted group: U = 0, $p$ < 0.001), and movement connotation scores were

higher for verbs, particularly the concrete ones, than for nouns (two-tailed Mann–Whitney tests, concrete nouns vs. concrete verbs in the blind group: U = 10, $p$ < 0.001; abstract nouns vs. abstract verbs in the blind group: U = 68,5, $p$ < 0.001; concrete nouns vs. concrete verbs in the sighted group: U = 6, $p$ < 0.001; abstract nouns vs. abstract verbs in the sighted group: U = 73,5, $p$ < 0.001) (see Supplementary Tables 3, 4 for average rating scores for all stimulus categories in the final stimulus list). Notably, there was a very high correlation between ratings provided by congenitally blind and sighted subjects on all three scales (two-tailed Pearson's correlations, concreteness ratings: r(94) = 0.96, 95% CI [0.94, 0.97], $p$ < 0.001; imaginability ratings: r(94) = 0.93, 95% CI [0.9, 0.96], $p$ < 0.001; movement connotation ratings: r(94) = 0.95, 95% CI [0.92, 0.97], $p$ < 0.001).

## fMRI experiment

In the fMRI experiment, the participants heard words and pseudowords in singular forms and were asked to mentally (i.e., without an overt response) transform them into plural forms, based on the verbal transformation cue presented beforehand ("many" and "few" for nouns, "we" and "they" for verbs). The subjects were explicitly instructed to treat pseudowords as the "real words" and produce transformations that sounded correct. Each word/pseudoword presentation lasted -0.5 s and was followed by 2 s of silence during which subjects were asked to create a correct word form in their minds. The time assigned for each trial was set at duration well above the average response times obtained in the pilot behavioral experiment (see Supplementary Table 2), to ensure that the participants were able to complete the task successfully.

The stimuli were presented in blocks of 6 items belonging to the same category (e.g., 6 abstract nouns), resulting in blocks lasting 15 s. The blocks were further grouped into "super blocks" according to the word grammatical class, such that 3 noun blocks (one with abstract nouns, one with concrete nouns, and one with pseudo nouns) were always grouped together and followed by 3 verb blocks, and vice versa. This second-level order was introduced to save time assigned for the presentation of transformation cues (i.e., the cue was the same for all blocks from the same grammatical class – thus, the introduction of the super blocks allowed us to present the cue only once per three blocks). Each super block started with a transformation cue, which was followed by 12 s of silence. Then, the three noun or three verb blocks were presented, each followed by 12 s of silence. Subsequently, the super block for the other grammatical class started. There were 4 noun super blocks and 4 verb super blocks in each fMRI run, resulting in presentation of 4 blocks for each stimulus category in each run. Subjects completed 4 runs, each lasting -12 min and 45 s. In one blind participant, the data collection during the 4th run was interrupted by an alarm and ensuing evacuation of the laboratory. In the case of this participant, the data from three runs were used in the analysis.

Each run involved presentation of different words and pseudowords (6 items per category per run). Thus, all between-run decoding analyses that we performed could not rely on word repetitions, and instead had to rely on a more abstract representation of linguistic properties of specific word/pseudoword classes. In each run, the same words were repeated in each block belonging to a specific experimental condition, but the presentation order was randomized. Furthermore, the block order, within each super block, was randomized with a constraint that the same order was applied to noun super blocks and verb super blocks.

In Polish, word grammatical classes have systematically different suffixes, which might result in certain analyses of differences between nouns and verbs being confounded by phonology. Furthermore, one can assume that the phonology of the verbal transformation cues, which subjects are likely to keep in mind during transformations, can confound these results. To be able to control for these issues, we systematically varied the phonology of nouns, verbs, and

transformation cues across the odd and even fMRI runs. In one type of runs we used WIELE ("many") and MY ("we") transformation cues for nouns and verbs, respectively, whereas in the other type of runs the transformation cues were KILKA ("few") and ONI ("they"). Different transformation cues resulted in different inflections for verbs, but not necessarily for nouns. Thus, in the case of nouns, one type of runs additionally included only masculine nouns, whereas the other type of runs included only feminine and neutral nouns. These manipulations resulted in a design in which any decoding analysis in an odd-even cross-validation scheme could not be driven by phonology. The run order was randomized across subjects, keeping the odd-even scheme in place.

The stimuli presentation was controlled by a program written in PsychoPy 3.0.12b[62]. The sounds were presented through MRI-compatible headphones. Before starting the experiment, each participant completed a short training session. Furthermore, the volume of sound presentation was individually adjusted to a loud, but comfortable level. The sighted participants were blindfolded for the duration of the fMRI experiment to create as similar environment of data acquisition for the sighted and the blind group as possible.

### Imaging parameters

Data were acquired on a 3-T Siemens Trio Tim MRI scanner using a 32-channel head coil at the Laboratory of Brain Imaging in Nencki Institute of Experimental Biology in Warsaw. Functional data were acquired using a multiband sequence with the following parameters: 60 slices, phase encoding direction from posterior to anterior; voxel size: 2.5 mm³; TR = 1.41 s; TE: 30.4 ms; multiband factor: 3. Before the start of the first functional run, T1-weighted anatomic scans were acquired using MPRAGE sequence with the following parameters: 208 slices, phase encoding direction from anterior to posterior; voxel size: 0.8 mm³; TR = 2.5 s; TE: 21.7 ms. The head of one blind participant could not fit a 32-channel head coil. Thus, this participant was scanned, with the same sequence parameters, using a larger, 12-channel head coil. The results obtained for this participant were not different from the results for other blind participants (see Source Data file, participant code: B17).

### MRI data analysis

**Data preprocessing.** The MRI data were converted from the DICOM format to the NIFTI format using the dcm2niix[63]. Then, the pre-processing was performed using SPM 12 (Wellcome Imaging Department, University College, London, UK, http://fil.ion.ucl.ac.uk/spm) and CONN 21b toolbox[64] running on MATLAB R2022a (MathWorks Inc. Natick, MA, USA). Data from each subject were preprocessed using the following routines: (1) functional realignment of all functional images; (2) direct segmentation and normalization, and (3) spatial smoothing with Gaussian kernel at 8-mm FWHM for the univariate analysis; no spatial smoothing for the multi-voxel pattern classification analysis.

Two first-level statistical models were created for each subject. For the multi-voxel pattern classification analysis, the data were modeled at the level of single blocks (24 predictors per run, one for each block). Additionally, transformation cues were modeled as conditions of no interest (8 predictors per run, one for each occurrence of the cue). For the univariate analysis, the data were modeled at the level of word categories (6 predictors per run, one for each word category, 1 predictor of no interest per run for cues). Signal time course was modeled using a general linear model by convolving a canonical hemodynamic response function with the time series of predictors. Six movement parameter regressors obtained during the preprocessing were added to the models. An inclusive high-pass filter was used (378 s, ~2 cycles per run) to remove drifts from the signal while ensuring that effects specific to each word category were not filtered out from the data. Autocorrelations were accounted for using autoregressive AR(1) model. Finally, individual beta maps, contrast maps, and t-maps were computed for each experimental block/condition, relative to rest periods.

**Multi-voxel pattern classification.** All multi-voxel pattern classification analyses were performed in CosmoMVPA[65] (v.1.1.0), running on Matlab R2022a (MathWorks). The analyses were performed on contrast maps for specific experimental blocks compared to rest periods (24 maps per run, 96 maps per participant in total). A linear support vector machine classification algorithm was used, as implemented in the LIBSVM toolbox[66] (v. 3.23). A standard LIBSVM data normalization procedure (i.e., Z-scoring beta estimates for each voxel in the training set and applying output values to the test set) was applied to the data before classification. The region of interest (ROI) analyses were performed using maps from the JuBrain Anatomy Toolbox[67] (v.3.0). Unless stated otherwise, the ROIs were defined bilaterally.

We first performed the omnibus ROI classification of activations for noun blocks and verb blocks in the visual areas, without dividing the stimuli into concrete, abstract, and pseudo words. The analysis included all occipital and occipitotemporal regions delineated in the JuBrain Anatomy Toolbox. To reduce a number of tests, subregions were combined (e.g., areas V3d and V3v were combined in area V3). The classification was performed in an odd-even cross validation scheme, that is, the classifier was trained on the odd runs and tested on the even runs, and vice versa, resulting in two cross validation folds. This scheme was used to ensure that the representation of phonological information did not affect the results (see also the "fMRI Experiment" section). The tests against classification chance level were corrected for multiple comparisons across the visual areas, within each group, using Bonferroni correction (correction for 8 tests). The between-group tests were performed only in areas in which significant results were observed in either group. Since we observed significant results only in area V5/MT in the blind participants, the correction for multiple comparisons was not necessary. Notably, the between-group difference reported in area V5/MT remained statistically significant even after Bonferroni correction across the visual areas (i.e., correction for 8 tests).

We further investigated the pattern of results in area V5/MT, in which the omnibus analysis showed significant effects. To this aim, we performed the classification of activations for noun blocks and verb blocks in this area, separately for each word category (concrete, abstract, pseudo). This more detailed analysis was performed in a leave-one-run-out cross validation scheme, that is, the classifier was iteratively trained on all runs except one and tested on the remaining run, resulting in four cross validation folds. This scheme let us test our hypothesis with maximal statistical power; at the same time, we expected positive results only for concrete word category, which means that the remaining word categories could serve as a control for phonological representation. The tests against classification chance level were corrected for multiple comparisons across the word categories, within each group, using Bonferroni correction (correction for 3 tests). Additionally, in each group, we performed a planned comparison between the classification results for concrete nouns and verbs and the average classification results for abstract and pseudo nouns and verbs (one test in each group, no correction for multiple comparisons necessary).

Next, we investigated whether topographic specificity of the effect reported in area V5/MT could be explained by the fact that its anatomical mask is smaller than the masks for other visual areas. To this aim, we drew the same number of voxels (193 voxels – the size of the V5 mask) from the mask of each visual region and reran the analyses described above in these subsets. For each participant, we performed 1000 random draws (without replacement) for each visual region and averaged the results across the draws. We then tested the averaged results against the classification chance level in a group analysis. The group tests were corrected for multiple comparisons,

using the Bonferroni correction, as was described above (i.e., correction for 8 tests in the initial omnibus analysis, correction for 3 tests in the analysis in specific semantic categories).

We used similar analytical approach to exclude the possibility that the results observed in area V5/MT in the blind participants were driven by only a small subset of voxels within this area's mask. We iteratively drew subsets of 40, 80, 120, and 160 voxels from the V5/MT mask and performed the classification of activity patterns for nouns and verbs from each semantic category (concrete, abstract, pseudo) in only these subsets. The voxels were drawn without replacements. For each participant, we performed 1000 iterations (i.e., 1000 random draws of specific voxels) of this analysis for each subset size, and we averaged the results across the iterations. We then tested the averaged results obtained at each analysis level against the chance level in a group analysis. The group tests at each analysis level were corrected for multiple comparisons across the three semantic categories, using the Bonferroni correction (correction for 3 tests).

Furthermore, we investigated how the removal of outliers influences the classification results obtained in area V5/MT for nouns and verbs from each semantic category. To this aim, we trimmed the results that diverged from the average classification accuracy for a given semantic category, in a given participant group, by more than 2 standard deviations. We then re-ran the group tests (i.e., the tests against classification chance level and the comparison across conditions) without these values. The results were corrected for multiple comparisons in the same way as in the main analysis.

We then performed the searchlight analysis to investigate which brain regions, beyond the visual cortex, capture differences between nouns and verbs in both participant groups. To this aim, we again used the omnibus classification approach, in which the experimental blocks were classified into noun blocks and verb blocks, without dividing the stimuli into concrete, abstract, and pseudo words. The classification was performed in volume space, in searchlight spheres with 5-voxel radius, using the odd-even cross validation scheme, similarly as the initial, omnibus ROI analysis. A statistical threshold for all tests performed on searchlight maps was set at $p < 0.001$ voxel-wise, corrected for multiple comparisons using family-wise error cluster (FWEc) correction approach.

We further studied the pattern of results in the superior temporal cortex, which showed the strongest effects in the searchlight analysis. We used a map of area TE3 from the JuBrain Anatomy Toolbox as a proxy of the superior temporal cortex, to define the ROI independently of the searchlight analysis. We then classified activity patterns for noun blocks and verb blocks in this region, separately in each word category. The tests against classification chance level were performed and corrected in the same way as the category-specific analysis in the visual area V5/MT. Additionally, in each group, we performed pairwise comparisons of the classification results for nouns and verbs across the three semantic categories. These tests were corrected for multiple comparisons using Bonferroni corrections (correction for 3 tests, in each group).

Furthermore, as a supplementary analysis, we also performed the classification of activity patterns for concrete word blocks and abstract word blocks. The classification was performed in a leave-one-run-out cross validation scheme, as the phonology of words was not systematically different across the concrete and abstract word categories. The classification was performed in both the visual ROIs and the superior temporal cortex ROI. The tests against classification chance level were corrected for multiple comparisons across the areas, within each group, using Bonferroni correction (correction for 9 tests). The between-group tests were performed only in areas in which significant results were observed in either group. Since we observed significant results only in areas V4 and V5 in the blind participants, the results of between-group comparisons were corrected for 2 tests, using Bonferroni correction.

Finally, two exploratory ROI analyses were performed. First, we tested whether the effect in area V5/MT was lateralized to the language-dominant hemisphere. Given that the lateralization of the language network is more variable in blind individuals than in sighted individuals[6,9,33–35], we empirically determined which hemisphere is language-dominant in each participant. To this aim, for each participant, we calculated a simple "lateralization index" of activations for language. We first averaged the activations for all words and pseudowords used in the study across three classic language regions: the left superior temporal cortex (Area TE 3 in the JuBrain Anatomy Toolbox), the left area 44, and the left area 45. We then subtracted the obtained value from the average activation calculated across the analogous regions in the right hemisphere. Thus, for each participant, the value greater than zero indicated left-lateralization of the activations for language in the language network, whereas the value lower than zero indicated right-lateralization.

As could be expected, our lateralization index indicated significant left-lateralization of activations for language in the sighted group (two-tailed, one-sample $t$-test, mean lateralization index value = 0.157, 95% CI [0.09, 0.22], t(19) = 4.88, $p < 0.001$, Cohen's d = 1.09). In contrast, no significant lateralization was detected in the blind group (two-tailed, one-sample $t$-test, mean lateralization index value = 0.059, 95% CI [−0.2, 0.14], t(19) = 1.54, $p = 0.14$, Cohen's d = 0.34). A direct comparison confirmed greater lateralization of activations for language in the sighted participants, compared to the blind participants (trend level, two-tailed, two-sample $t$-test, mean difference = 0.1, 95% CI [0, 0.2], t(38) = 1.97, $p = 0.056$, Cohen's d = 0.62). The lateralization indices showed that, in 6 out of 20 blind participants, linguistic stimuli activated the right analogs of classic language regions more strongly. In sighted participants, only 2 out of 20 participants showed this pattern of results. This is in line with reports that, in the typical population, language is left-lateralized in ~92% of individuals[68,69], and shows that our lateralization measure worked as expected.

We used individual language lateralization indices to run the analyses – both the omnibus classification of activations for all nouns and all verbs and the more detailed analysis of activations for nouns and verbs from specific semantic categories - in area V5/MT in the language dominant hemisphere and in the language non-dominant hemisphere separately, in each participant. The group tests against chance classification level were corrected within each group using Bonferroni correction (correction for 4 tests: one test in the omnibus analysis and three tests – one for each semantic category - in the detailed analysis). No significant results were detected in either group – thus, no between-group comparisons were run.

We also further investigated whether differences between nouns and verbs were represented in the high-level ventral and ventrolateral visual areas. To this aim, we ran our analyses - both the omnibus classification of activations for all nouns and all verbs and the more detailed analysis of activations for nouns and verbs from specific semantic categories - in the lateral occipitotemporal cortex (LOTC) and ventral occipitotemporal cortex (VOTC). The LOTC mask was defined as a sphere with 10-mm radius centered on a peak of preferential responses to words that refer to tools, relative to other semantic categories, in blind and sighted individuals, as reported by Peelen et al.[35] (Talairach coordinates −50 −60 −5, transformed to MNI coordinates −53 −60 −12 using BioImage Suite - https://bioimagesuiteweb.github.io/webapp/mni2tal.html). An analogous sphere was also created in the right hemisphere (MNI coordinates of the center: 53 −60 −12), and then the masks in both hemispheres were merged in order to create a bilateral ROI. The bilateral VOTC mask was taken from the study by Mattioni et al.[36], which reported sound-induced categorical effects in this region in blind and sighted individuals. The group tests against the classification chance level were corrected for multiple comparisons across the word categories,

separately for each region and group, using the Bonferroni correction (correction for 4 tests). In the omnibus analysis, the between group comparison was run only for the VOTC, in which significant effect was observed in the blind group (no correction for multiple comparisons necessary). In the detailed analysis, the only significant effect – above-chance classification of activations for pseudo nouns and verbs in the VOTC in the blind participants – was compared to the results for other semantic categories, obtained in the same region and group (Bonferroni correction for 2 tests).

In the ROI classification analyses, the statistical significance of obtained classification accuracies was tested against chance levels that were empirically derived in the permutation procedure. Specifically, each classification analysis was re-run 1000 times for each participant with the labels of classified conditions randomly assigned to experimental blocks in each iteration. Null distributions created in this procedure were averaged across participants and compared with the actual average classification accuracies. The $p$ values that were obtained in this way were corrected for multiple comparisons, as was described above. A review of null distributions confirmed that, for each ROI and analysis, the empirically-derived chance levels were indistinguishable from a priori chance levels (50%). Thus, for simplicity, the a priori chance level is presented in the figures. The analyses of effects in specific subsets of voxels drawn from the anatomical masks (Supplementary Figs. 3–5, 7) were already based on permutations (see above). Thus, in these specific analyses, using the permutation procedure for significance testing was not practical, and testing against classification chance level was performed with one-tailed, one-sample $t$-tests.

Testing for differences in classification accuracy between the participant groups was performed with two-tailed, two-sample $t$-tests. Testing for differences in classification accuracy between the two conditions was performed with two-tailed, paired $t$-tests. Testing for significant interactions between the results for ROIs, conditions, and participant groups was performed with mixed ANOVAs, with the results of post-hoc tests corrected for multiple comparisons using Bonferroni correction. SPSS 25 (IBM Corp, Armonk, NY) was used to perform these tests.

In the searchlight analysis, the individual classification results were entered into SPM group models. SPM one-tailed, one-sample $t$-tests were used to compare the results of each searchlight analysis with chance level, separately in the blind and the sighted group. SPM one-tailed, two-sample $t$-tests were used to compare the results between groups.

**Univariate analysis.** We first performed the whole-brain univariate analysis, in which we compared the activation induced by all words and pseudowords to rest periods, in both participant groups. The activations for all experimental conditions, relative to rest periods, were averaged at the single-subject level. Then, the average activation maps were entered into SPM one-tailed, one-sample $t$-tests, performed separately for each group, and into the SPM one-tailed, two-sample $t$-tests, which tested for the between-group differences. As in the searchlight classification analysis, the statistical threshold for these analyses was set at $p < 0.001$ voxel-wise, corrected for multiple comparisons using FWEc correction approach.

The whole-brain analysis was followed by the ROI analysis in the visual cortex. The analysis of activations for all experimental conditions, relative to rest periods, was performed in the same visual ROIs that were used in the multi-voxel pattern classification analysis. Then, two-tailed, one-sample $t$-tests were used to compare the activations for all conditions with activations during rest periods, separately in the blind and the sighted group. Bonferroni correction was used to correct the results for multiple comparisons across all ROIs used in the analysis, for each participant group separately (correction across 8 tests). Two-tailed, two-sample $t$-tests were then used to compare the results between groups, within each ROI that showed significant results in at least one participant group. The results of these tests were corrected using Bonferroni correction (correction across 8 tests).

Finally, we performed the ROI analysis of activations for each experimental condition, relative to rest periods, in the area V5/MT. Two-tailed, one-sample $t$-tests were again used to compare the activations for each condition with activations during rest periods, separately in the blind and the sighted group. Bonferroni correction was used to correct the results across all conditions, for each participant group separately (correction across 6 tests). The differences across experimental conditions and groups were tested with mixed ANOVA. The results of the post-hoc tests were corrected using Bonferroni correction. SPSS 25 was used to perform all statistical tests in the univariate ROI analysis.

**Data visualization.** The results of ROI analyses were visualized using Matlab R2022a and the DataViz toolbox (https://github.com/povilaskarvelis/DataViz). The results of whole-brain analyses were visualized using BrainNet Viewer[70]. The masks of brain areas used in the ROI analyses (Supplementary Figs. 1 and 11) were visualized using MRIcroGL (https://www.nitrc.org/projects/mricrogl).

#### Reporting summary
Further information on research design is available in the Nature Portfolio Reporting Summary linked to this article.

### Data availability
The neuroimaging data generated in this study have been deposited at the Open Science Framework (https://osf.io/vqwuk/). Source data are provided with this paper.

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

## Acknowledgements

This work was supported by a National Science Center Poland grant (2020/37/B/HS6/01269) and a Polish National Center for Academic Exchange fellowship (BPN/SEL/2021/1/00004) awarded to Ł.B.

## Author contributions

M.U., A.C. and Ł.B. conceptualized and designed the study; M.U. and M.P. collected the data; M.U. and Ł.B. performed the data analyses; M.U. and Ł.B. wrote the manuscript; A.C. revised the manuscript.

## Competing interests

The authors declare no competing interests.
