## [Transparent Peer Review file · Nature Communications]

Neural representation of nouns and verbs in congenitally blind and sighted individuals

Corresponding Author: Dr Lukasz Bola

Version 0:

Reviewer comments:

Reviewer #1

(Remarks to the Author)

Plasticity of the brain in visually deprived or blind animals and humans has been a prime research theme for over 50 years, but it keeps giving. The present manuscript deals exclusively with humans (barely mentioning animal models!) and is specifically asking how nouns and verbs are represented in the brains of congenitally blind individuals. The research question of whether and how language is represented in the visual cortex of blind people has been controversial for some time. The present report provides a relatively clear answer: yes, elements of language (nouns and verbs) are found to be represented in areas of visual cortex in the blind, but not in primary regions and only with highly specific properties.

The results are noteworthy in two ways: 1) they confirm that elements of language are indeed represented in sensory regions of the blind brain, but 2) they refute earlier suggestions of a pluripotent cortex that can turn into anything. Braille reading, which one may call at least borderline linguistic, may be largely explained through affine sensory transformation from one sensory modality to another. By contrast, the claim of “real” linguistic processing in a purely sensory region like primary visual cortex has always seemed silly to me. The authors’ idea that the brain must use related computational principles and transformations to “compute physical and spatial representations of the world” appears much closer to reality.

Nevertheless, several questions remain that I would like the authors to answer:

1) Having established the principle of “computational equivalency” between sensory and language domains, how is V5, a classical area in the visual dorsal stream primarily for the processing of motion, computationally equivalent to (or capable of) representing certain types of language elements, such as nouns or verbs? That question is not sufficiently answered.

2) Similarly, why did the authors decide to concentrate on V5, which is a key area of the visual dorsal stream, often thought to be responsible specifically for the processing of motion? The explanation given in the text (“linguistic effects in the blind visual cortex are driven by, or are an extension of, typical visuospatial computations performed in this region”) does not sound fully convincing to me. Other areas in the dorsal stream, like V3, in the middle occipital gyrus (MOG), play a more direct (and broader) role in spatial vision in the sighted and are tuned to motion in space in the blind (Renier et al., 2010). Furthermore, it seems from Fig. 1 that the results in some of the ventral-stream areas (V6, LO, FG) come close to significance as well, and one could argue that this makes them eligible too. (It also makes sense that the lower-level areas are far from significance.) The way to answer this question, it seems to me, is that each of these other areas performs a specific computation, which takes place in both sighted and in blind.

For instance, it was recently shown that the fusiform face area (FFA) is activated in the blind by auditory substitutions of face elements, so the computational purpose of each area remains the same (Plaza et al., 2023). This supports the notion that a specific cortical region performs the same computational operation on the incoming input regardless of its sensory modality. The authors of the present study seem to entertain a similar conclusion in the last sentence of the Discussion, but they could elaborate on this more.

3) Repeatedly, the authors use the term ‘canonical language region’ for the superior temporal cortex (STC). Scholars of auditory cortical processing will almost certainly take offense with this characterization. Undoubtedly, STC performs language functions in a hierarchical fashion along the auditory ventral stream (phonemes/syllables to words to expressions/phrases), as shown by DeWitt & Rauschecker (2012). But equally clearly STC participates in much wider

functions than just language processing, including various forms of auditory processing, music, and a variety of other complex sounds. Processing of speech sounds is just one of many functions that STC performs, so calling it a 'canonical language region' seems somewhat misleading.

4) Use of the above terminology, mistaking the STC primarily for a language region, is, in my opinion, one of the overall shortcomings of this study. It seems as if the authors consider language a parallel universe rather than something made up of real-world sensory stimuli. The authors would be well advised to rethink their results and consider, perhaps even re-interpret them, from a 'complex-audition' point of view. It seems that the argument they are trying to make with their data would be much easier to make if they framed it in these terms.

5) More minor is the fact that the term V5/MT is (appropriately) amalgamated from the European V5 (Zeki) and the American MT (Allman & Kaas, 1971). All of these citations deserve to be acknowledged, if and when that term is used.

6) The idea of reverse hierarchies, as mentioned by the authors, could be useful.

7) The summary paragraph at the end of the Discussion is again excellent and could perhaps be expanded by taking into account analogies of operation in the visual and auditory domain. A discussion of phonology could also be considered.

Reviewer #2

(Remarks to the Author)

I enjoyed reading the paper of Urbaniak et al. The question authors set-up to investigate -what is encoded in the occipital cortex of blind people when they process words- is interesting and timely. Overall the paper is well-written, the design is sound and the analyses carried out are mostly adequate (but see below).

I however also think that the paper does not reach its full potential by omitting some specific brain regions, analyses and contextualization of the results in a broader literature.

I think that the authors do not fully support their claim with what seems to me the obvious analyses to do based on the data they have: using Representational Similarity Analyses to assess more directly whether "movement" is the encoded feature in V5. Indeed, contrasting noun and verb, in addition to contrast different movement connotation also contrast different classes of linguistic stimuli (this is collinear) and potentially different behavioral difficulties in the task at play (not directly accessible here since no overt response recorded) etc.... A more sensitive way to address this might be to create movement model expressed as dissimilarity matrices using stimuli classes as control models (even if some correlations are expected between models, this can be regressed). This is particularly suitable here since some verbs appear to have low movement (eg to talk, to hear) while some nouns might have high movement connotation (eg chicken, hand).

It is not clear from the method whether the authors did corrections for multiple comparisons taking the number of ROIs into account? This is particularly important given that the searchlight results do not confirm the ROI analyses (no occipital regions show enhanced decoding for verb vs noun in the blind; Fig. 3). How do the authors explain these discrepancies, which might be a major limitation to the robustness of the results.

The decision not to include ROIs from the ventral occipital-temporal stream is puzzling given the proposition these regions involve in linguistic/categorical coding in blind people. I think it would be relevant to include them.

Since the known laterality of language functions, including in the occipital cortex of blind people (Bedny), it would be interesting to see if the V5 effect laterals to the left by not merging ROIs from L and R hemispheres.

Why Movement connotation express solely at the multivariate level and not in the univariate analyses could be further developed. What does that tell us about the way the regions encode "implied" movement.

The absence of any significant decoding in V1 for any comparisons in blind people is puzzling given previous demonstration that V1 can decode sound -which also activate specific lexical entries- (Vetter; Mattioni) and words (Seydell-Greewald) categories. On the later study, I have to admit I don't understand the discussion of the authors on the discrepancies across studies.

I think the authors should be more straightforward in the abstract, intro, discussion etc... about what they mean by " These finding suggest that the blind visual cortex represents the physical properties of nouns and verbs....".

Please be explicit by what you mean by "physical properties ". Here authors test movement connotation. Does that mean that ANY other linguistic properties map onto ANY other occipital regions in blind? Surely this was not tested and such a statement is at odd with some literature so the phrasing could be more explicit.

Reviewer #3

(Remarks to the Author)

This is a clearly written report of a nicely designed study of the processing of (nouns vs verbs) x (concrete vs abstract vs

pseudo) in the brain of blind vs typical participants. The core claim of the authors is that MVPA in area V5 allows to decode nouns vs verbs, in the blind but not in the control group. My main concern is that this result appears statistically rather weak. Reading through the Results section, I noted the following points:

- L 148: I appreciated that in the "omnibus analysis" Bonferroni correction was applied. Still I noted that contrary to what is stated in the methods not all occipitotemporal regions of the JuBrain atlas were apparently included (the LOC was omitted).
- L 152 I suspect that there is some "double dipping" in the (critical) comparison between groups of decoding in V5. Indeed, V5 was first selected by virtue of its high level of decodability in the blind, which then biases the subsequent comparison of decodability in the blind minus in controls.
- L 157: I guess that it is a just a consequence of the different decoding designs in the two cases, and I know that comparing p values is not the right way to go, but is it not surprising that decoding nouns vs verbs seems to be working less well when decoding was restricted to the supposedly decodable concrete items ($p=0.024$) than when it was applied to the whole set of data including abstract and pseudo items?
- L 163: consistent with my remark on line 152, this more stringent ANOVA shows no significant group difference in decodability in V5
- L 179: I regretted that the separate results of the R and L hemispheres were never reported, particularly for V5 of course. In the field of language, pooling a priori the L and R ROIs without even looking into their differences is a bit unusual.
- L 180: one may regret that V5 was not identified individually with some movement localizer, as the location of V5 varies notably across individuals, and maybe more so between blind and typical individuals (doi: 10.1093/cercor/bhw180 <https://doi.org/10.1016/j.cogbrainres.2005.08.015>).
- L 189: the searchlight analysis did not confirm the results of the ROI analysis in V5
- L 200-206: the authors report nice results, without however further supporting the existence of a significant group difference in V5.
- L 217: the decoding of concrete vs abstract words, which was predicted to "accompany" the decoding of concrete verbs vs nouns, was significant in V4 and V5, but with a group difference restricted to V4, and not V5.

Version 1:

Reviewer comments:

Reviewer #2

(Remarks to the Author)

The authors should be praised for their thorough and thoughtful reply.

I find the clarification that the noun-verb decoding is selective to the concrete grammatical class and selective only to the V5/MT ROI convincing.

I only have a few remaining points of clarification.

One thing that remains unclear is that the region of interest analyses were performed using maps from the JuBrain Anatomy Toolbox (Eickhoff et al., 2005). All ROIs were defined bilaterally. However all those regions have different sizes and therefore a different number of voxels as features for the decoder. How does that impact the results?

In an additional analysis, classification of activity patterns for nouns and verbs from specific semantic categories was performed in subsets of voxels from the V5/MT mask. These subsets of voxels (40, 80, 120, 160 voxels) were iteratively drawn from the V5/MT mask. Were those different voxel sets randomly chosen from the bilateral ROIs? This is surprising this works even with 40 voxels if taken from a bilateral mask, while it does not work with a unilateral mask including much more voxels.

Related to that point, the fact that representations of concrete verb vs sounds rely on bilateral networks is intriguing. I am however not fully convinced by the argument provided by the authors. The fact that the process recruits both hemispheres is different from the fact that decoding only works if patterns of activity include both hemispheres. For instance, motion direction is encoded in left and right V5/MT but directions can be decoded in each of them separately. Why would the process necessarily engage a representation that requires both hemispheres to be reliably decoded?

Authors suggest that responses to language in the blind visual cortex can be driven by relatively low-level, spatial and/or physical representations. This suggests that the visual cortex receives physical properties of word referents rather than more abstract, conceptual properties of word referents or linguistic properties of words. I sympathize with this idea. But how do the authors privilege this idea rather than the possibility that occipital regions implement semantic representations stored as abstract knowledge independent of simulating the physical features of what a specific word refers to? I understand it is selective for motion but could it be an "abstract" representation of implied motion rather than the simulation of physical motion in the visual (sighted) or tactile, auditory (blind) modalities?

Given that classification is significant only between concrete nouns and verbs; what is the meaning of Fig. 1 which, if I am not mistaken, mix the different grammatical classes ?

Reviewer #3

(Remarks to the Author)

The authors made commendable efforts to address the points I raised in the first round of reviews. They confirm my observation that the differences between blind and seeing participants are not entirely water tight, which, although the theoretical discussion accommodates this state of affairs, reduces somewhat the originality of the work. Regarding my observation on “double dipping”, the authors provide two answers. I believe that the first one is not quite correct, in as much as they DO select V5 based on significant decoding in the blind. It would not be double dipping if it had been selected based on decoding accuracy in the average of both groups. They provide a second, better response, based on the fact that the group difference was significant in V5 even when Bonferroni-corrected, something which they should put in the paper.

Reviewer #1 (Remarks to the Author):

Plasticity of the brain in visually deprived or blind animals and humans has been a prime research theme for over 50 years, but it keeps giving. The present manuscript deals exclusively with humans (barely mentioning animal models!) and is specifically asking how nouns and verbs are represented in the brains of congenitally blind individuals. The research question of whether and how language is represented in the visual cortex of blind people has been controversial for some time. The present report provides a relatively clear answer: yes, elements of language (nouns and verbs) are found to be represented in areas of visual cortex in the blind, but not in primary regions and only with highly specific properties.

The results are noteworthy in two ways: 1) they confirm that elements of language are indeed represented in sensory regions of the blind brain, but 2) they refute earlier suggestions of a pluripotent cortex that can turn into anything. Braille reading, which one may call at least borderline linguistic, may be largely explained through affine sensory transformation from one sensory modality to another. By contrast, the claim of “real” linguistic processing in a purely sensory region like primary visual cortex has always seemed silly to me. The authors’ idea that the brain must use related computational principles and transformations to “compute physical and spatial representations of the world” appears much closer to reality.

We would like to thank the reviewer for this positive feedback. In response to specific comments, we strived to further clarify our hypothesis, the logic of our study, and our interpretation of the results.

We think that the focus of our manuscript on humans is a natural consequence of our research question. Our primary aim was to disentangle semantic and grammatical representations in the visual cortex of blind individuals. While we agree that broader review of research on the plasticity of the blind visual cortex, both in humans and animals, would be useful for some readers, we are already over the word limit set by *Nature Communications*.

That said, we refer to studies on animal models that showed weakening of inhibitory pathways in the blind visual cortex, a result that was a direct inspiration for the ideas expressed in the paper (Benevento et al., 1995; Morales et al., 2002). We now also cite the work of Allman & Kaas on the middle temporal gyrus in the owl monkey, as suggested by the Reviewer (Allman and Kaas, 1971). We would welcome the Reviewer’s suggestions about any other studies on animal models that are directly relevant to our work.

Nevertheless, several questions remain that I would like the authors to answer:

1) Having established the principle of “computational equivalency” between sensory and language domains, how is V5, a classical area in the visual dorsal stream primarily for the processing of motion, computationally equivalent to (or capable of) representing certain types of language elements, such as nouns or verbs? That question is not sufficiently answered.

We now elaborate on this issue in the manuscript. We clarify that our study originates from a simple, but often overlooked fact that words are highly multidimensional objects. A given brain area can represent differences between words because of how they sound (phonological dimension), what they mean (semantic dimension), or what role they play in a sentence (grammatical dimension). The aim of our study was to investigate which word properties are represented in the visual cortex of blind individuals

Our finding that the visual area V5/MT in blind individuals represents differences between concrete nouns and verbs, but not between abstract or pseudo nouns and verbs, cannot be explained by auditory representations, as the phonological properties of words were not systematically different across these semantic categories. Similarly, this pattern of results cannot be driven by grammatical representations – in all semantic categories, words were readily recognizable as nouns or verbs. Thus, these results suggest that area V5/MT in blind individuals represents differences between nouns and verbs through their semantic representations, that is, through representations of objects and actions named by words.

Our study further shows that area V5/MT in blind individuals capture properties that are saliently and systematically different for concrete noun and verb referents, but not necessarily for abstract noun and verb referents. We hypothesized that the most plausible candidate for such a property is representation of motion connotations, which is likely differently activated by concrete nouns, generally naming stationary objects, and concrete verbs, generally naming dynamic actions. As the reviewer mentioned, area V5/MT primarily processes motion. It is known to be sensitive not only to visual, but also to auditory motion (Poirier et al., 2005, Rezk et al., 2020), with the auditory sensitivity being preserved and elevated in blind individuals (Bedny et al., 2010; Strnad et al., 2013; Dormal et al., 2016). Our study suggests that this area can also represent motion connotations of objects and actions that are retrieved from semantic representations. In other words, our findings indicate that this area can retrieve the same physical property from memory-driven semantic representations and stimulus-driven perceptual (visual, auditory, etc.) representations. This is the computational equivalency that can explain why we are finding representation of differences between concrete nouns and verbs - word classes with starkly different motion connotations - primarily in the motion-sensitive area V5/MT.

To better explain these crucial issues, we now expanded the introduction (lines 78-96):

“We enrolled congenitally blind and sighted participants in a functional magnetic resonance imaging (fMRI) experiment, in which they made morphological transformations (singular-to-plural number transformations) to spoken concrete nouns and verbs, abstract nouns and verbs, and morphologically-marked pseudo nouns and verbs. The words were chosen based on behavioral ratings by blind and sighted participants, and task difficulty was equated across nouns and verbs (see Methods and Supplementary Tables 1-4). The study used a block design (always presenting 6 words from the same category in a row) and different words were presented in each fMRI run. We used multi-voxel pattern classification to reveal brain areas representing the noun/verb distinction across and within the three semantic categories (concrete, abstract, pseudo) in both participant groups.

This design allowed us to disentangle grammatical and semantic representations in the visual cortex of blind individuals. Specifically, if the blind visual cortex represents differences between nouns and verbs at the grammatical level, we should observe comparable classification of activity patterns for nouns and verbs from all semantic categories in this region - in all categories, words were readily recognizable as nouns or verbs. Conversely, modulation of results by semantic category would suggest that the blind visual cortex represents differences between nouns and verbs at the semantic as opposed to the grammatical (morphosyntactic) level. To investigate these two possibilities, we controlled for phonological effects in all key analyses (see Methods).”

Then, in the discussion, we write (lines 341-370):

“Words are highly multidimensional objects. A given brain area can represent differences between words because of how they sound (phonological dimension), what they mean (semantic dimension), or what role they play in a sentence (grammatical dimension). The aim of our study was to investigate which word properties are represented in the visual cortex of blind individuals, and could drive activation for linguistic stimuli in this region⁵⁻¹⁰. We focused on a fundamental linguistic distinction, that between nouns and verbs, and investigated whether semantic or grammatical aspects of this distinction are represented in the blind visual cortex. We found above-chance classification of activity patterns for nouns and verbs in visual area V5/MT in the blind participants, but not in other visual areas in this group. We further showed that the effect in area V5/MT in the blind was primarily driven by successful classification of activations for concrete nouns and verbs, in the absence of significant results for abstract and pseudo nouns and verbs. Different classification results for nouns and verbs from different semantic categories cannot be explained by differences in auditory representations, as the phonological properties of words were not systematically different across the semantic categories. Similarly, this pattern of results cannot be driven by grammatical representations – in all semantic categories, words were readily recognizable as nouns or verbs. Thus, these results suggest that area V5/MT in blind individuals represents differences between nouns and verbs because of their differing semantic properties, that is, through representations of objects and actions named by words.

Our study further shows that area V5/MT in blind individuals captures properties that are saliently and systematically different for concrete noun and verb referents, but not necessarily for abstract noun and verb referents. We suggest that the most plausible candidate for such a property is the representation of motion connotations, which is likely differently activated by concrete nouns, generally naming stationary objects, and concrete verbs, generally naming dynamic actions. Area V5/MT is sensitive to visual²³ and auditory^{24,25} motion, with auditory sensitivity being preserved and elevated in blind individuals²⁶⁻²⁸. Our study suggests that this area can also retrieve motion connotations of objects and actions from semantic representations. In other words, our findings indicate that this area can retrieve the same physical property from memory-driven semantic representations and stimulus-driven perceptual (visual, auditory, etc.) representations.”

2) Similarly, why did the authors decide to concentrate on V5, which is a key area of the visual dorsal stream, often thought to be responsible specifically for the processing of motion? The explanation given in the text (“linguistic effects in the blind visual cortex are driven by, or are an extension of, typical visuospatial computations performed in this region”) does not sound fully convincing to me. Other areas in the dorsal stream, like V3, in the middle occipital gyrus (MOG), play a more direct (and broader) role in spatial vision in the sighted and are tuned to motion in space in the blind (Renier et al., 2010). Furthermore, it seems from Fig. 1 that the results in some of the ventral-stream areas (V6, LO, FG) come close to significance as well, and one could argue that this makes them eligible too. (It also makes sense that the lower-level areas are far from significance.) The way to answer this question, it seems to me, is that each of these other areas performs a specific computation, which takes place in both sighted and in blind.

For instance, it was recently shown that the fusiform face area (FFA) is activated in the blind by auditory substitutions of face elements, so the computational purpose of each area remains the same (Plaza et al., 2023). This supports the notion that a specific cortical region performs the same computational operation on the incoming input regardless of its sensory

modality. The authors of the present study seem to entertain a similar conclusion in the last sentence of the Discussion, but they could elaborate on this more.

We agree with the theoretical perspective described by the Reviewer. Our study suggests that specific visual areas can retrieve the same type of information not only from perceptual experience in different sensory modalities (perception of faces in either the visual modality or the auditory modality activates the FFA), but also from the semantic representations computed in higher-level brain regions (hearing a word “John” activates the semantic representation of a person I know, the features of his face are retrieved from the memory and back projected to the FFA, which results in activation of this region). We now explain this in the revised discussion. In addition to the paragraphs cited above, we also write (lines 389-418):

“Our findings suggest that, after the activation of semantic representations in higher-level brain regions, the physical properties of word referents, retrieved from these representations, are back projected to the visual system in a way that parallels feedforward visual processing. Thus, the “motion template” of objects and actions is back projected to the dorsal stream areas, such as area V5/MT, the “shape template” is back projected to the ventral stream areas, and so on. Such organization of the back projections from the semantic system might be most useful for forming visual predictions and, consequently, supporting visual perception⁴⁰. This mechanism might be preserved and functional in blind individuals, even if its original function is not relevant in this population. Moreover, the impact of such modulations on the visual cortex activity might be greater in blindness because of the lack of competition from feedforward visual inputs and weakening of inhibitory mechanisms in the visual areas in this population^{41,42}. This view implies that our results were restricted to area V5/MT not because this area is especially sensitive to semantic properties of words, compared to other visual areas. Instead, the stronger results in this area were observed simply because we classified activity patterns for nouns and verbs, that is, two word classes with differing motion connotations. Other contrasts should reveal representations of other physical properties of word referents in other visual areas in blind individuals. Furthermore, based on this hypothesis, qualitatively similar, even if weaker results should be expected in sighted individuals. In line with this prediction, our analysis revealed statistical trends in area V5/MT in the sighted participants (Supplementary Fig. 4), which are qualitatively similar to the results observed in this area in the blind participants.

This view about activations induced in the blind visual cortex by language processing concurs with findings in the perceptual domain. Research in this domain shows that the processing of auditory and tactile stimuli by blind individuals activates specific visual areas that, in sighted individuals, process comparable stimuli in the visual modality⁴³⁻⁴⁵. Some of these auditory and tactile effects were also detected in the visual cortex of sighted individuals^{24,25,46-48}. Based on these results, it has been suggested that many visual areas can retrieve comparable information from perceptual experiences in different sensory modalities. Here, we show that this “computational equivalency” principle may also organize back projections from higher-level semantic regions to the visual cortex.”

Finally, we would like to note that while we have concentrated on area V5/MT in the main text - we hope that the revised manuscript better explains why we expected to find the positive results primarily in this area - we did not constrain our analysis to this area. In the supplementary materials, we present exactly the same analyses for all other visual areas

included in the study (see Supplementary Fig. 2). We also describe these analyses in the main text (lines 167-179):

“As a control analysis, we investigated the accuracy of classification of activity patterns for nouns and verbs for each word category in all visual areas considered in the study (Supplementary Fig. 2). This analysis was meant to search for meaningful effects that were specific to only one word category and could have been potentially missed in the initial, omnibus analysis. However, apart from the already reported effect in area V5/MT in the blind participants, we only observed above-chance classification of activations for pseudo nouns and verbs in several visual areas in the blind participants. This result could perhaps be driven by some form of surprise response to these atypical stimuli. Critically, this effect was not accompanied by the successful classification of activations for abstract nouns and verbs in any of the visual areas (all p values > 0.25). Furthermore, besides the already reported effect in area V5/MT, the classification of activations for concrete nouns and verbs did not produce significant results in any other visual area (all p values > 0.25). This confirms the topographic specificity of the effect reported for concrete nouns and verbs in area V5/MT.”

3) Repeatedly, the authors use the term ‘canonical language region’ for the superior temporal cortex (STC). Scholars of auditory cortical processing will almost certainly take offense with this characterization. Undoubtedly, STC performs language functions in a hierarchical fashion along the auditory ventral stream (phonemes/syllables to words to expressions/phrases), as shown by DeWitt & Rauschecker (2012). But equally clearly STC participates in much wider functions than just language processing, including various forms of auditory processing, music, and a variety of other complex sounds. Processing of speech sounds is just one of many functions that STC performs, so calling it a ‘canonical language region’ seems somewhat misleading.

As we now clarify in the manuscript, we never meant that the STC is involved only in language. We termed it a “canonical language region” (we have now changed it to the “classic language region”) only because it has been shown to be involved in language processing in hundreds of studies, in participants from various cultures and using different languages. This does not exclude the involvement of this region in many other tasks.

Even in the context of language processing, we believe that it is important to remember that STC can represent different kinds of information and serve different functions. One function is perceptual processing of speech, as the reviewer nicely described. However, the STC is also a reservoir of more abstract semantic and grammatical representations, which are not necessarily tied to the auditory properties of the signal themselves. We believe that this conclusion is well supported by converging evidence from many research lines (see recent review by Fedorenko et al., 2024, for a number of relevant references).

This second linguistic function of the STC is also evident in our study. Specifically, auditory processing could not affect the classification of activity patterns for nouns and verbs in our searchlight analysis, as phonology of words was systematically altered across the odd and the even runs, and the classification was performed in an odd/even cross-validation scheme (we elaborate on this in the Methods). Thus, the results observed in the superior temporal cortex in the searchlight analysis must be driven by linguistic representations (semantic, grammatical, etc.) that are independent of phonological processing.

We now explain these points in the Discussion, in lines 481-498:

“Beyond the visual cortex, we observed the representation of differences between nouns and verbs only in the classic language regions, in both the blind and the sighted participants. In both groups, this representation was strongest in the middle and superior temporal cortices. This region is involved in perceptual analysis of auditory signals, which also includes parsing speech into phonemes, words, and expressions⁵⁸. However, besides perceptual processing of speech, the superior temporal cortex also computes semantic and grammatical representations, which are largely independent of auditory properties of speech themselves³. In our study, auditory processing could not affect the classification of activity patterns for nouns and verbs in the searchlight analysis, as phonology of words was systematically altered across the odd and even runs, and the classification was performed in the odd/even cross-validation scheme (see Methods). Thus, the results observed in the searchlight analysis in the superior temporal cortex must be driven by linguistic representations that are independent of phonological processing. Apart from the superior temporal cortex, there were several, additional effects in other classic language regions in the blind participants. One possible explanation is that blind individuals are more attentive to speech⁵⁹, which can lead to more robust signal in certain language areas. Despite these between-group differences, our results suggest that the overall topography of the classic language network is robust to changes in visual experience.”

4) Use of the above terminology, mistaking the STC primarily for a language region, is, in my opinion, one of the overall shortcomings of this study. It seems as if the authors consider language a parallel universe rather than something made up of real-world sensory stimuli. The authors would be well advised to rethink their results and consider, perhaps even re-interpret them, from a ‘complex-audition’ point of view. It seems that the argument they are trying to make with their data would be much easier to make if they framed it in these terms.

We probably would not use the term “parallel universe”, but we do think that, at the level of semantic and grammatical representations, language processing is largely abstracted away from the sensory properties of input. The word “butterfly” will activate the representation of the same animal no matter if written, spoken, or signed. This word will remain a noun and will serve the same function in a sentence in all these sensory modalities. A large number of studies indicate that such sensory-independent linguistic representations are primarily stored in the “core” language regions, such as the STC, inferior frontal gyrus, and anterior temporal lobe. These representations seem to be largely independent from auditory representations necessary for speech comprehension in the STC, and motoric representations necessary for speech production in the inferior frontal gyrus (reviewed in: Fedorenko et al., 2024).

As we clarify in responses above and in the revised manuscript (e.g., lines 78-96 and 341-359), we believe that auditory processing cannot explain our key finding that the visual area V5/MT in blind individuals represents differences between concrete nouns and verbs, but not between abstract or pseudo nouns and verbs (Fig. 2). The phonological properties of words were not systematically different across these semantic categories. Similarly, auditory processing cannot explain the results of the omnibus analyses, implemented in the form of either the ROI analysis (Fig. 1) or the searchlight analysis (Fig. 3). In these analyses, the classification of activity patterns for nouns and verbs was performed in the odd-even cross-validation scheme, and we systematically varied the phonology of words used in the odd and the even runs. Thus, the classification algorithm could not rely on phonological information.

We hope that the revisions we have made (see our responses to comments above) clarified these important issues.

5) More minor is the fact that the term V5/MT is (appropriately) amalgamated from the European V5 (Zeki) and the American MT (Allman & Kaas, 1971). All of these citations deserve to be acknowledged, if and when that term is used.

We corrected this in the manuscript, thank you.

6) The idea of reverse hierarchies, as mentioned by the authors, could be useful.

We now describe in the manuscript how the “reverse flow” of information, through back projections from the semantic regions, can explain our results in the visual cortex. As was cited above, in lines 389-409 we now write:

“Our findings suggest that, after the activation of semantic representations in higher-level brain regions, the physical properties of word referents, retrieved from these representations, are back projected to the visual system in a way that parallels feedforward visual processing. Thus, the “motion template” of objects and actions is back projected to the dorsal stream areas, such as area V5/MT, the “shape template” is back projected to the ventral stream areas, and so on. Such organization of the back projections from the semantic system might be most useful for forming visual predictions and, consequently, supporting visual perception⁴⁰. This mechanism might be preserved and functional in blind individuals, even if its original function is not relevant in this population. Moreover, the impact of such modulations on the visual cortex activity might be greater in blindness because of the lack of competition from feedforward visual inputs and weakening of inhibitory mechanisms in the visual areas in this population^{41,42}. This view implies that our results were restricted to area V5/MT not because this area is especially sensitive to semantic properties of words, compared to other visual areas. Instead, the stronger results in this area were observed simply because we classified activity patterns for nouns and verbs, that is, two word classes with differing motion connotations. Other contrasts should reveal representations of other physical properties of word referents in other visual areas in blind individuals. Furthermore, based on this hypothesis, qualitatively similar, even if weaker results should be expected in sighted individuals. In line with this prediction, our analysis revealed statistical trends in area V5/MT in the sighted participants (Supplementary Fig. 4), which are qualitatively similar to the results observed in this area in the blind participants.”

7) The summary paragraph at the end of the Discussion is again excellent and could perhaps be expanded by taking into account analogies of operation in the visual and auditory domain. A discussion of phonology could also be considered.

In response to the reviewer's comments, we significantly expanded the whole Discussion section. As described above, we now clarify, throughout the manuscript (e.g., lines 78-96 and 341-359), why auditory and phonological processing of word forms could not explain our key results.

We want to reiterate that we do not question that certain properties of objects, recognized either through hearing or through touch, can be mapped directly onto the visual cortex in a way that respects the “computational equivalency” between perceptual processing streams in different sensory modalities. As the Reviewer pointed out, there are many studies showing that this can be the case. However, as we explain above, such a “direct route” between sensory systems cannot explain the effects reported in our study, particularly the fact that area V5/MT in blind individuals represents differences between concrete nouns and verbs, but not between abstract or pseudo nouns and verbs. There is nothing in the auditory form of

the words used in our study that indicates whether they refer to concrete objects and actions or abstract concepts and actions.

We argue that the effects observed in area V5/MT can be only explained by an indirect, semantic route, in which (1) words are first processed in the auditory perceptual stream, which leads to their recognition and activation of semantic representations downstream. Then, (2) certain semantic properties (i.e., physical properties of objects and actions), activated downstream, are back projected to the visual cortex in a manner that parallels the flow of feedforward visual inputs. Our study specifically suggests that motion connotations of objects and actions - the physical property that is differentially captured by concrete nouns and verbs - is back projected to the motion-sensitive area V5/MT.

We now clarify these points in lines 410-418 of the revised manuscript (already cited in response to previous comments):

“This view about activations induced in the blind visual cortex by language processing concurs with findings in the perceptual domain. Research in this domain shows that the processing of auditory and tactile stimuli by blind individuals activates specific visual areas that, in sighted individuals, process comparable stimuli in the visual modality⁴³⁻⁴⁵. Some of these auditory and tactile effects were also detected in the visual cortex of sighted individuals^{24,25,46-48}. Based on these results, it has been suggested that many visual areas can retrieve comparable information from perceptual experiences in different sensory modalities. Here, we show that this “computational equivalency” principle may also organize back projections from higher-level semantic regions to the visual cortex.”

We thank the Reviewer for encouraging us to elaborate these important issues.

Reviewer #2 (Remarks to the Author):

I enjoyed reading the paper of Urbaniak et al. The question authors set-up to investigate - what is encoded in the occipital cortex of blind people when they process words- is interesting and timely. Overall the paper is well-written, the design is sound and the analyses carried out are mostly adequate (but see below).

I however also think that the paper does not reach its full potential by omitting some specific brain regions, analyses and contextualization of the results in a broader literature.

We thank the reviewer for this encouraging feedback. In response to the reviewer’s specific comments, we made a number of revisions and clarifications, which are described below. We also present several new analyses, which investigate the robustness of our key findings and adds broader context to our study.

I think that the authors do not fully support their claim with what seems to me the obvious analyses to do based on the data they have: using Representational Similarity Analyses to assess more directly whether "movement" is the encoded feature in V5. Indeed, contrasting noun and verb, in addition to contrast different movement connotation also contrast different classes of linguistic stimuli (this is collinear) and potentially different behavioral difficulties in the task at play (not directly accessible here since no overt response recorded) etc.... A more sensitive way to address this might be to create movement model expressed as dissimilarity matrices using stimuli classes as control models (even if some correlations are expected between models, this can be regressed). This is particularly suitable here since some verbs appear to have low movement (eg to talk, to hear) while some nouns might have high movement connotation (eg chicken, hand).

We agree that RSA would be an interesting way to further investigate the reported effect in area V5/MT. However, as we now clarify right away in the Introduction (lines 83-85), we (1) used a block design (always presenting 6 words from the same category in a row) and (2) presented different words in each fMRI run.

The block design was used to maximize statistical power in our planned comparisons in the multi-voxel pattern classification analysis, since we wanted to study differences in activations induced by various word classes in the visual cortex, and we reasoned that such effects cannot be expected to be very strong, particularly in sighted individuals. The presentation of different words in each run was used to ensure that our investigation, primarily focused on semantic and grammatical representations, was not confounded by relatively trivial factors, such as domain-general (e.g., attentional) or phonological activations driven by repetitions of the same words across runs (we used cross validation across runs in all classification analyses).

While we believe that these features of our study worked well for the classification analyses we planned, they also make a proper RSA analysis impossible to perform.

We decided to optimize our design for multi-voxel pattern classification, rather than for RSA, because our main aim was to disentangle semantic and grammatical representations of nouns and verbs in the visual cortex of blind individuals. We believe that RSA would not give these two alternatives the same, fair chances. Specifically, the RSA model of grammatical differences would be a purely categorical model, which does not predict differences between words from specific categories. Such a model would be very different and likely less powerful than the RSA model of semantic differences, which would predict variability not only across categories but also across individual words, within each category. In the classification analysis, in contrast, we always classified the two categories of words.

That said, we believe that confounding factors listed by the reviewer are well controlled in our study, even without RSA. Controlling for these factors was actually the reason why we studied the representation of differences between nouns and verbs from the three semantic categories (concrete, abstract, and pseudo words).

- 1) First, the representation of grammatical class (i.e., the representation of different classes of linguistic stimuli, as the Reviewer described it) cannot explain our results in area V5/MT, because the classification of activity patterns for nouns and verbs was successful only in the concrete words category. There were no significant results for abstract and pseudo nouns and verbs, even though the words in these categories (particularly in the abstract category) were as readily recognizable as nouns and verbs as were the words in the concrete category.
- 2) Second, our results in area V5/MT cannot be explained by task-related or attentional confounds, because the task demands were the same for the concrete and abstract words - yet, the results for these two semantic categories were different. The morphological transformations that the participants were asked to perform during fMRI were identical for concrete words and for abstract words. It is true that we did not collect behavioral data in the fMRI task (asking participants to produce overt responses would likely induce too much head motion). However, we performed an additional behavioral study, in which we asked the participants to perform the same task but produce an overt response. This study showed that response times for

concrete and abstract words are statistically indistinguishable (see Table S1), confirming our claim of same task demands in the two semantic categories.

- 3) Third, similar analysis can be applied to assess potential influence of phonological representations on the results in area V5/MT. Auditory or phonological processing cannot explain our results because the phonological properties of words were not systematically different across the semantic categories.

Overall, our key result in area V5/MT cannot be explained by grammatical, phonological, or domain-general (attentional or related to task difficulty) factors. Thus, we suggest that area V5/MT in blind individuals represents differences between nouns and verbs through semantic representations, that is, through representations of objects and actions named by words. Only differences in semantic representation can explain different classification results for nouns and verbs for the three semantic categories used in our study.

Our study further shows that area V5/MT in blind individuals captures properties that are saliently and systematically different for concrete noun and verb referents, but not necessarily for abstract noun and verb referents. We hypothesized that the most plausible candidate for such a property is representation of motion connotations, which are likely differently activated by concrete nouns, generally naming stationary objects, and concrete verbs, generally naming dynamic actions. Area V5/MT is sensitive to visual (Zeki et al., 1991) and auditory (Poirier et al., 2005, Rezk et al., 2020) motion, with the auditory sensitivity being preserved and elevated in blind individuals (Bedny et al., 2010; Strnad et al., 2013; Dormal et al., 2016). Our study suggests that this area can also represent motion connotations of objects and actions that are retrieved from semantic representations. In other words, our findings indicate that this area can retrieve the same physical property from memory-driven semantic representations and stimulus-driven perceptual (visual, auditory, etc.) representations.

We now strived to better explain this logic of our study in the Introduction, in lines 78-96:

“We enrolled congenitally blind and sighted participants in a functional magnetic resonance imaging (fMRI) experiment, in which they made morphological transformations (singular-to-plural number transformations) to spoken concrete nouns and verbs, abstract nouns and verbs, and morphologically-marked pseudo nouns and verbs. The words were chosen based on behavioral ratings by blind and sighted participants, and task difficulty was equated across nouns and verbs (see Methods and Supplementary Tables 1-4). The study used a block design (always presenting 6 words from the same category in a row) and different words were presented in each fMRI run. We used multi-voxel pattern classification to reveal brain areas representing the noun/verb distinction across and within the three semantic categories (concrete, abstract, pseudo) in both participant groups.

This design allowed us to disentangle grammatical and semantic representations in the visual cortex of blind individuals. Specifically, if the blind visual cortex represents differences between nouns and verbs at the grammatical level, we should observe comparable classification of activity patterns for nouns and verbs from all semantic categories in this region - in all categories, words were readily recognizable as nouns or verbs. Conversely, modulation of results by semantic category would suggest that the blind visual cortex represents differences between nouns and verbs at the semantic as opposed to the grammatical (morphosyntactic) level. To investigate these two possibilities, we controlled for phonological effects in all key analyses (see Methods).”

We further clarify these critical points in the Discussion, in lines 341-370:

“Words are highly multidimensional objects. A given brain area can represent differences between words because of how they sound (phonological dimension), what they mean (semantic dimension), or what role they play in a sentence (grammatical dimension). The aim of our study was to investigate which word properties are represented in the visual cortex of blind individuals, and could drive activation for linguistic stimuli in this region⁵⁻¹⁰. We focused on a fundamental linguistic distinction, that between nouns and verbs, and investigated whether semantic or grammatical aspects of this distinction are represented in the blind visual cortex. We found above-chance classification of activity patterns for nouns and verbs in visual area V5/MT in the blind participants, but not in other visual areas in this group. We further showed that the effect in area V5/MT in the blind was primarily driven by successful classification of activations for concrete nouns and verbs, in the absence of significant results for abstract and pseudo nouns and verbs. Different classification results for nouns and verbs from different semantic categories cannot be explained by differences in auditory representations, as the phonological properties of words were not systematically different across the semantic categories. Similarly, this pattern of results cannot be driven by grammatical representations – in all semantic categories, words were readily recognizable as nouns or verbs. Thus, these results suggest that area V5/MT in blind individuals represents differences between nouns and verbs because of their differing semantic properties, that is, through representations of objects and actions named by words.

Our study further shows that area V5/MT in blind individuals captures properties that are saliently and systematically different for concrete noun and verb referents, but not necessarily for abstract noun and verb referents. We suggest that the most plausible candidate for such a property is the representation of motion connotations, which is likely differently activated by concrete nouns, generally naming stationary objects, and concrete verbs, generally naming dynamic actions. Area V5/MT is sensitive to visual²³ and auditory^{24,25} motion, with auditory sensitivity being preserved and elevated in blind individuals²⁶⁻²⁸. Our study suggests that this area can also retrieve motion connotations of objects and actions from semantic representations. In other words, our findings indicate that this area can retrieve the same physical property from memory-driven semantic representations and stimulus-driven perceptual (visual, auditory, etc.) representations.”

We now also acknowledge that lack of RSA could be seen as limitation of our study (lines 499-508):

“Two limitations of our work should be acknowledged. First, the searchlight analysis did not confirm the results obtained in area V5/MT in the analysis in visual regions of interest. This difference in results across these two methods is likely driven by the fact that the searchlight analysis included V5/MT voxels from only one hemisphere at a time. Our analyses show that only including activity patterns from both hemispheres results in robust effects. Second, we did not confirm that area V5/MT represents motion connotations of word referents with representational similarity analysis. Our study was optimized for multi-voxel pattern classification and used a block design and different words in each experimental run. While these methodological choices served an important purpose in our study, they also precluded the representational similarity analysis.”

It is not clear from the method whether the authors did corrections for multiple comparisons taking the number of ROIs into account? This is particularly important given that the searchlight results do not confirm the ROI analyses (no occipital regions show enhanced

decoding for verb vs noun in the blind; Fig. 3). How do the authors explain these discrepancies, which might be a major limitation to the robustness of the results.

As we now clarify in the Methods and the figure captions:

- 1) In the initial analysis in the visual cortex (Fig. 1), the tests against classification chance level were corrected across ROIs, within each group (Bonferroni correction across 8 tests). The between-group tests were performed only in areas in which significant results were observed in either group. Since we observed significant results only in area V5/MT in the blind participants, the correction for multiple comparisons was not necessary. Incidentally, the between group comparison in this area was strong enough ($p = 0.006$) to survive the Bonferroni correction across ROIs.
- 2) In the more detailed analysis in area V5 (Fig. 2), the tests against classification chance level were corrected across the semantic categories (concrete, abstract, pseudo), within each group (Bonferroni correction across 3 tests). The same correction logic was used in the analogous analyses in other visual regions (Supplementary Fig. 2) and in the superior temporal cortex (Fig. 4)

Thus, the p values for our critical results in area V5/MT in blind individuals were:

- 1) Above-chance classification of all nouns and verbs in the initial, omnibus analysis (Fig. 1): $p = 0.001$ (corrected $p = 0.008$)
- 2) Above-chance classification of concrete nouns and verbs in the more detailed analysis (Fig. 2): $p = 0.008$ (corrected $p = 0.024$). Furthermore, we now show that removing the outliers from the data further strengthens this result (new $p = 0.001$, corrected $p = 0.003$)
- 3) Comparison between classification accuracy obtained for concrete nouns and verbs and the average classification accuracy for abstract and pseudo nouns and verbs: $p = 0.023$ (one test for a participant group, no correction necessary). We now show that removing the outliers from the data further strengthens this result (new $p = 0.004$)

We believe that these are statistically robust effects, which show that area V5/MT in blind individuals preferentially represents differences between concrete nouns and verbs, relative to the representation of differences between abstract and pseudo nouns and verbs. As we now clarify in the manuscript, we believe that this is our key result. While the differences between groups are less robust, we do not expect them to be robust. In fact, our hypothesis specifically predicts that linguistic effects in the blind visual cortex are driven by uncovering and strengthening of mechanisms present also in the sighted brain, rather than by more dramatic, qualitative changes in computed representations. In support of this hypothesis, we now show that removal of outliers from the data reveals statistical trends in the area V5/MT in the sighted group, which are qualitatively similar to the results found in this area in the blind group (see below).

As the Reviewer noted, the searchlight analysis did not confirm the results obtained for area V5/MT in the analysis in visual regions of interest. This difference in results across the two methods is likely driven by the fact that the searchlight analysis included voxels from only one hemisphere at a time. As we now show, in response to the Reviewer's other comment, only the analysis performed on activity patterns from both hemispheres results in robust effects.

Notably, we now report two additional analyses, which further underscore the robustness of results obtained in area V5/MT in the blind participants when the activity patterns in both hemispheres are analyzed jointly. First, we wanted to exclude that the observed results are driven by only a small subset of voxels within the V5/MT mask. This analysis is described in lines 180-194:

“We investigated the robustness of our results in area V5/MT with two additional analyses. First, we wanted to rule out that the observed results are driven by only a small subset of voxels within the V5/MT mask – for example, only those voxels that border more anterior areas that represent actions at increasingly conceptual level³². To investigate this possibility, we iteratively drew subsets of voxels (40, 80, 120, and 160 voxels, 1000 random draws at each level) from the V5/MT mask and performed the classification of activations for nouns and verbs from each semantic category in only these subsets (Supplementary Fig. 3). We found above-chance classification of activations for concrete nouns and verbs in blind individuals across all analysis levels (p value for the analysis on 40 voxels = 0.068; all other p values < 0.05). Thus, this effect does not depend on several specific voxels within the V5/MT mask, and can be reliably detected across a broad spectrum of analysis parameters. In contrast, no significant effects, at any analysis level, were observed for abstract and pseudo nouns and verbs (all p values > 0.25). As in the main analysis, no significant effects for nouns and verbs from any semantic category were observed in the sighted group (all p values > 0.25).”

Second, we reviewed the robustness of the results at the level of individual data. As mentioned above, we ensured that our findings are not driven by outliers. This analysis is described in lines 195-223:

“Second, we reviewed the classification results obtained in area V5/MT for individual participants to ensure that our findings were not driven by outliers (Supplementary Fig. 4). In blind individuals, the above-chance classification of activations for concrete nouns and verbs in this area was observed in 14 out of 20 participants (70% of the participants). Across the results for three semantic categories in the blind group, 4 values were identified as potential outliers, defined as observations diverging from average classification accuracy in a given category by more than 2 standard deviations (1 value in the concrete category, 1 value in the abstract category, and 2 values in the pseudo category). Removing these values from the analysis further strengthened the group effects. Particularly, we still found above-chance classification of activations for concrete nouns and verbs (an increase in average classification accuracy from 55.9% to 57.2%, $p = 0.003$), but not for abstract and pseudo nouns and verbs (both p values > 0.25). In the direct comparison across the categories we again found that the classification accuracy obtained for concrete nouns and verbs was significantly higher than the average classification accuracy calculated across the abstract and pseudo nouns and verbs (mean difference = 7.6%, 95% CI [2.8%, 12.3%], $t(15) = 3.36$, $p = 0.004$, Cohen’s $d = 0.84$). In the sighted group, using the same data trimming procedure resulted in 2 values (1 value in the concrete category, 1 value in the abstract category) being removed from the data. Interestingly, after this procedure we found an uncorrected effect for concrete nouns and verbs also in area V5/MT in the sighted participants (an increase in average classification accuracy from 53% to 54.1%, uncorrected $p = 0.037$), in the absence of effects for the abstract and pseudo nouns and verbs (both uncorrected $p > 0.25$). The above-chance classification of activations for concrete nouns and verbs in this area was observed in 13 out of 20 sighted participants (13 out of 19 participants after data trimming,

68% of the participants). The direct comparison across semantic categories indicated a trend toward a higher classification accuracy for concrete nouns and verbs, compared to the average classification accuracy for abstract and pseudo nouns and verbs (mean difference = 6.3%, 95% CI [-0.2%, 12.7%], $t(18) = 2.05$, $p = 0.056$, Cohen's $d = 0.48$). Overall, the analysis of trimmed data suggests that, while statistically weaker, the results in area V5/MT in the sighted group might be qualitatively similar to those found in the blind group.”

Overall, we believe that our key results are robust. The discrepancy between the ROI analysis and the searchlight analysis likely reflects “ground truth” - the fact that the former analysis is performed on combined data from both hemispheres, whereas the latter on the data from only one hemisphere at a time. We elaborate on this issue in our responses below.

The decision not to include ROIs from the ventral occipital-temporal stream is puzzling given the proposition these regions involve in linguistic/categorical coding in blind people. I think it would be relevant to include them.

We apologize if this was unclear, but we did include the fusiform gyrus ROI (FG, combination of areas FG 1-4 from the JuBrain Anatomy Toolbox), which encompassed most of the ventral occipitotemporal cortex. In the figure below, we present the overlap between FG ROI and the VOTC ROI taken from a recent study (Mattioni et al., 2020) that investigated sound-induced categorical representations in this region in blind and sighted individuals. While the VOTC ROI used by Mattioni and colleagues is clearly larger than the FG ROI, the overlap between these two ROIs is also quite clear. To avoid any further confusion, we now visualize the location of all ROIs used in our study in the supplementary materials (see new Supplementary Fig. 1 in the revised manuscript)

Figure R1. Comparison of the FG ROI (red), used in our study, and the VOTC ROI (green) used by Mattioni et al. (2020).

Using the larger VOTC ROI from the study by Mattioni and colleagues, instead of the FG ROI from the JuBrain Anatomy Toolbox, produces slightly stronger results in the omnibus classification of all nouns and all verbs in the blind group (FG, corrected $p = 0.064$; VOTC, corrected $p = 0.048$ if the same correction across 8 ROIs is used; corrected $p = 0.024$ when the correction across conditions, within the VOTC, is applied: see the Methods section in the revised manuscript). However, the more detailed analysis suggested that this effect is driven primarily by successful classification of pseudo nouns and verbs (corrected $p = 0.02$), in the

absence of any significant results for concrete and abstract nouns and verbs (both uncorrected $p > 0.25$). We did not find any significant effects in the VOTC in the sighted participants.(all corrected $p > 0.2$).

Besides the analysis in the VOTC mask, we now also report the analysis in the lateral occipitotemporal complex (LOTC), for completeness (only more posterior, lateral occipital area - LO - was investigated in the original manuscript). We defined this region functionally, as a 10-mm sphere centered on a peak of preferential responses to words that refer to tools, relative to other semantic categories, in blind and sighted individuals, as reported by Peelen et al. (2013; Talairach coordinates: -50 -60 -5; these coordinates were transformed to MNI coordinates -53 -60 -12 using BiImage Suite - <https://bioimagesuiteweb.github.io/webapp/mni2tal.html>). However, we did not detect any significant effects in the LOTC, in either the omnibus or the more detailed analysis (all corrected p values in the blind group > 0.25 , all corrected values in the sighted group > 0.05).

We now describe these analyses in the new section “exploratory analyses”, in lines 317-329:

“We also further investigated whether differences between nouns and verbs were represented in high-level ventral and ventrolateral visual areas. To this aim, we ran our analyses in the lateral occipitotemporal cortex (LOTC) and the ventral occipitotemporal cortex (VOTC), as defined by previous studies that reported categorical effects in these regions in blind individuals^{36,37} (Supplementary Fig. 8). In the omnibus classification of activity patterns for all nouns and verbs, we found a significant effect in the VOTC in the blind group (mean classification accuracy = 53,7%, $p = 0.024$). However, the more detailed analysis suggested that this effect is driven primarily by successful classification of pseudo nouns and verbs (mean classification accuracy = 55,9%, $p = 0.02$), in the absence of significant results for concrete and abstract nouns and verbs (both $p > 0.25$). We did not find robust effects in the LOTC in either group (all p values > 0.1) or in the VOTC in the sighted group (all p values > 0.25). Overall, the differences between concrete or abstract nouns and verbs do not seem to be robustly represented in ventral or ventrolateral visual areas.”

Since the known laterality of language functions, including in the occipital cortex of blind people (Bedny), it would be interesting to see if the V5 effect lateralizes to the left by not merging ROIs from L and R hemispheres.

We believe that what Bedny and coauthors repeatedly showed is that, at the group level, lateralization of activations for language is less clear in blind individuals than in sighted individuals. This effect is primarily driven by a greater interindividual variability in the blind group - a significant portion of blind individuals actually show comparable activations for linguistic stimuli in both hemispheres, or even stronger activations in the right hemisphere (Lane et al., 2015; Lane et al., 2017; see also Röder et al., 2000, 2002; Dziegiel-Fivet and Jednoróg, 2024). Thus, testing the hypothesis that the effects in our study are stronger in the “language-dominant” hemisphere first requires empirically determining what hemisphere it is in each participant - especially in each blind participant. Using the left hemisphere as “language-dominant” could induce a bias in comparisons between sighted and blind individuals, as that would capture the language lateralization better in the sighted group than in the blind group.

We tested if these lateralization differences can be detected also in our data. For each participant, we calculated a simple “lateralization index” of activations for language. We first

averaged the activations for all words and pseudowords used in the study across three classic language regions: the left superior temporal cortex (Area TE 3 in the JuBrain Anatomy Toolbox), the left BA 44, and the left BA 45. We then subtracted the obtained value from the average activation calculated across the analogous regions in the right hemisphere. Thus, for each participant, the value greater than 0 indicated left-lateralization of the activations for language in the language network, whereas the value lower than 0 indicated right-lateralization.

Indeed, our lateralization index indicated significant left-lateralization of activations for language in the sighted group (mean lateralization index value = 0.157, $t = 4.89$, $p < 0.001$). In contrast, no significant lateralization was detected in the blind group (mean lateralization index value = 0.059, $t = 1.54$, $p = 0.14$). A direct comparison confirmed greater lateralization of activations for language in the sighted participants, compared to the blind participants (trend level, $t = 1.97$, $p = 0.056$). The lateralization indices showed that, in 6 out of 20 blind participants, linguistic stimuli activated the right analogs of classic language regions more strongly. In sighted participants, only 2 out of 20 participants showed this pattern of results. This is in line with reports that, in the typical population, language is left-lateralized in approximately 92 % of individuals (results from right-handers only: Knecht et al., 2000; results from both right-handers and left-handers: Labache et al., 2023), and suggests that our basic lateralization measure is actually quite accurate. We describe these sanity checks in the Methods section of the revised manuscript (lines 764-774).

We used individual language lateralization indices to run, in each participant, the analysis in area V5/MT in the language dominant hemisphere and in the language non-dominant hemisphere separately. However, the results obtained in this analysis were weaker than those produced by the bilateral analysis. We now report these results in lines 298-316:

“We tested whether the effects observed in the classification analysis in area V5/MT were lateralized to the language-dominant hemisphere. Given that the lateralization of the language network is more variable in blind individuals than in sighted individuals^{6,9,33-35}, we empirically determined which hemisphere is language-dominant in each participant by comparing the magnitudes of activations for words and pseudowords in classic language regions (superior temporal and inferior frontal cortices) and in their analogs in the right hemisphere (see Methods). For each participant, we then ran separate classification analysis in area V5/MT in the language-dominant and the language-nondominant hemisphere. However, the group effects obtained in these analyses (Supplementary Fig. 7) were weaker than those produced by the bilateral analysis. In the language-dominant hemisphere, we observed uncorrected effects in the omnibus classification of activity patterns for all nouns and all verbs in both blind individuals (mean classification accuracy = 52.3%, uncorrected $p = 0.038$) and sighted individuals (mean classification accuracy = 52%, uncorrected $p = 0.049$). However, these results did not survive the correction for multiple comparisons (both p values > 0.15). Furthermore, no significant results, in either group and hemisphere, were detected in the more detailed analysis, in which we classified activity patterns for nouns and verbs from each semantic category separately (all p values > 0.1). This suggests that the robust results reported in area V5/MT in blind individuals in the main analysis were driven by activity patterns in both hemispheres.”

As we now discuss, this analysis further challenges the account that, in the absence of visual inputs, the visual cortex is “colonized” (Bedny, 2017) by signals from the language areas and, in essence, becomes a part of the language network. In that case, one could

expect the results to be stronger in the language-dominant hemisphere. In the manuscript, we develop a different theoretical perspective - we propose that, in sighted individuals, the physical properties of word referents are back projected to the visual system in order to prepare the visual system for the likely incoming stimulation (i.e., to predict incoming visual information; Rao and Ballard, 1999). When you hear “watch out for the ball” it is likely that the ball will enter your visual field in a moment, and it might be adaptive to prepare the visual system for a quick detection of this object. We propose that this mechanism is preserved in blind individuals and, combined with weakening of inhibitory mechanisms in the blind visual cortex, might drive strong responses to language in this region.

Based on our theoretical perspective, it is only expected that differences between concrete nouns and verbs in area V5 of blind individuals are comparably represented in the language-dominant and the language non-dominant hemisphere, and that, consequently, the results are stronger in the bilateral analysis. If the original goal of this mechanism was to support vision, then this information should be projected to both hemispheres

We now explain these important points in the Discussion (lines 389-434):

“Our findings suggest that, after the activation of semantic representations in higher-level brain regions, the physical properties of word referents, retrieved from these representations, are back projected to the visual system in a way that parallels feedforward visual processing. Thus, the “motion template” of objects and actions is back projected to the dorsal stream areas, such as area V5/MT, the “shape template” is back projected to the ventral stream areas, and so on. Such organization of the back projections from the semantic system might be most useful for forming visual predictions and, consequently, supporting visual perception⁴⁰. This mechanism might be preserved and functional in blind individuals, even if its original function is not relevant in this population. Moreover, the impact of such modulations on the visual cortex activity might be greater in blindness because of the lack of competition from feedforward visual inputs and weakening of inhibitory mechanisms in the visual areas in this population^{41,42}. This view implies that our results were restricted to area V5/MT not because this area is especially sensitive to semantic properties of words, compared to other visual areas. Instead, the stronger results in this area were observed simply because we classified activity patterns for nouns and verbs, that is, two word classes with differing motion connotations. Other contrasts should reveal representations of other physical properties of word referents in other visual areas in blind individuals. Furthermore, based on this hypothesis, qualitatively similar, even if weaker results should be expected in sighted individuals. In line with this prediction, our analysis revealed statistical trends in area V5/MT in the sighted participants (Supplementary Fig. 4), which are qualitatively similar to the results observed in this area in the blind participants.

This view about activations induced in the blind visual cortex by language processing concurs with findings in the perceptual domain. Research in this domain shows that the processing of auditory and tactile stimuli by blind individuals activates specific visual areas that, in sighted individuals, process comparable stimuli in the visual modality⁴³⁻⁴⁵. Some of these auditory and tactile effects were also detected in the visual cortex of sighted individuals^{24,25,46-48}. Based on these results, it has been suggested that many visual areas can retrieve comparable information from perceptual experiences in different sensory modalities. Here, we show that this “computational equivalency” principle may also organize back projections from higher-level semantic regions to the visual cortex.

Furthermore, the above-described view is well aligned not only with our key findings, but also with several supplementary results we report. First, we found that the representation of differences between nouns and verbs in area V5/MT in blind individuals is not lateralized to the language-dominant hemisphere. Such lateralization can be expected based on the account that, in the absence of visual inputs, visual cortex is “colonized”⁴⁹ by signals from the language areas and, in essence, becomes a part of the language network. However, in the conjecture described above we argue that the original function of the back projections from the semantic system to the visual cortex is to support visual perception. In that case, the information carried by these back projections should reach visual areas in both hemispheres and the lateralization should not necessarily be expected. Second, the broad contrast between all concrete words (with physical referents) and all abstract words (without physical referents) produced results in both dorsal and ventral visual areas in blind individuals, and even in the primary visual cortex (Supplementary Fig. 5). Our hypothesis predicts these areas are sensitive to specific physical properties of objects and actions conveyed by concrete words – those properties that are typically represented in these areas during feedforward visual processing.”

Why Movement connotation express solely at the multivariate level and not in the univariate analyses could be further developed. What does that tell us about the way the regions encode "implied" movement.

We now extend the discussion of this aspect of our results by referring the reader to several other studies, which used non-linguistic stimuli, but reported similar observation in the blind visual cortex - that is, detection of an effect of interest in the multivariate, but not the univariate analysis (Vetter et al., 2020; Bola et al., 2023). We believe that this adds important context to our results, because it shows that the difference between multivariate and univariate effects is not specific to linguistic stimuli that we used in our study. It seems that this is a more general phenomenon. Thus, we do not think that it tells us something specifically about how linguistic stimuli are represented in the blind visual cortex.

We now discuss two possible explanations of this difference. It can be entirely driven by differences in power of these two analysis modes. However, it is also possible that more complex effects (such as semantically-driven effects in our study) are truly represented in the blind visual cortex at the level of large-scale activity patterns, which cannot be detected even by the most powerful univariate (i.e., voxel-by-voxel) analysis.

We discuss these possibilities in lines 445-464 of the revised manuscript:

“The univariate analyses reported here showed activations for spoken words in the visual cortex in blind participants, a result that is consistent with previous studies. Interestingly, the analysis in area V5/MT in this group suggests that the global magnitude of activation in this region, as measured by univariate methods, captures different processes than the multi-voxel activation patterns. The global activation of area V5/MT in the blind was the same for all word classes, but higher for pseudowords, which suggests that this measure captured increased processing demands related to surprising or atypical stimuli. Attention or cognitive load modulates the activation magnitude in the visual cortex even in sighted individuals^{50,51}. The increase in sensitivity of the blind visual cortex might strengthen these effects, leading to the generic response that can cover semantically-driven processes, specific for a given word category. These more specific processes can nevertheless be detected with multivariate analysis. One possibility is that the difference in results obtained with these two analytical approaches is driven entirely by differences in statistical power. However, it is also possible

that more complex effects are truly represented at the level of large-scale activity patterns, which cannot be detected even by the most powerful univariate (i.e., voxel-by-voxel) analysis. Notably, such disparity in results of these two analytical approaches, with the multivariate analysis showing results that were not detected in the univariate analysis, were reported in several recent studies of the blind visual cortex that did not use linguistic stimuli^{52,53}. Thus, this may be a more general phenomenon, not specifically tied to linguistic processing.”

The absence of any significant decoding in V1 for any comparisons in blind people is puzzling given previous demonstration that V1 can decode sound -which also activate specific lexical entries- (Vetter; Mattioni) and words (Seydell-Greewald) categories. On the later study, I have to admit I don't understand the discussion of the authors on the discrepancies across studies.

The classification of activity patterns for all concrete words and all abstract words (Supplementary Fig. 5) in area V1 was actually significant at the uncorrected level (uncorrected $p = 0.012$). The fact that the effect in area V1 in this analysis did not survive our conservative correction for multiple comparisons (Bonferroni correction across 9 areas tested) is perhaps not surprising given that our design was not optimized for this supplementary contrast, and, consequently, we used only 2/3 of the collected data in this analysis. We now report the uncorrected effects observed in this analysis in V1 and in other visual areas in lines 256-269:

“Finally, the hypothesis that the blind visual cortex is sensitive to words because it represents the physical features of word referents implies that this region captures differences not only between concrete nouns and verbs, but also between concrete and abstract words. While our design was not optimized for this contrast, we tested this prediction and found a robust, above-chance classification of activity patterns for concrete and abstract words in areas V4 (mean classification accuracy = 54.6%, $p = 0.018$) and V5 (mean classification accuracy = 54.7%, $p = 0.027$) in the blind participants (Supplementary Fig. 5). Additionally, the effects in areas V1, V6, the lateral occipital area, and the fusiform gyrus in the blind group were significant at the uncorrected level (all uncorrected p values < 0.05), but did not survive the correction for multiple comparisons. In contrast, the classification of activations for concrete and abstract words was not successful in the superior temporal cortex in this group, even at the uncorrected level (uncorrected $p = 0.24$) (Supplementary Fig. 5). This further suggests that the effects observed in this classic language region might be driven by different neural representation than the effects found in the blind visual cortex.”

Overall, our results suggest that area V1 might represent differences between all concrete words and all abstract words (uncorrected $p = 0.012$), but it does not represent differences between nouns and verbs (see Fig. 1 and Supplementary Fig. 2). This suggests that this low-level visual area can be sensitive to certain, but definitely not all semantic dimensions conveyed by words.

This is in line with our theoretical approach. As we now clarify in the manuscript, we do not think that area V5/MT is especially sensitive to semantic properties of words, compared to other visual areas. We believe that the effect detected in our study was present primarily in area V5/MT because our contrast - between nouns, naming generally stationary objects, and verbs, naming generally dynamic actions - matched the typical predilection of this area to process motion, not only from visual inputs, but also from other sources (auditory and tactile

experience and, as our study suggests, also semantic representations). Other contrasts, capturing other physical dimensions, should reveal representation of these dimensions in other visual regions. Perhaps some of these representations are projected back even to area V1.

We clarify these points in lines 389-434 (already cited in responses to previous comments):

“Our findings suggest that, after the activation of semantic representations in higher-level brain regions, the physical properties of word referents, retrieved from these representations, are back projected to the visual system in a way that parallels feedforward visual processing. Thus, the “motion template” of objects and actions is back projected to the dorsal stream areas, such as area V5/MT, the “shape template” is back projected to the ventral stream areas, and so on. Such organization of the back projections from the semantic system might be most useful for forming visual predictions and, consequently, supporting visual perception⁴⁰. This mechanism might be preserved and functional in blind individuals, even if its original function is not relevant in this population. Moreover, the impact of such modulations on the visual cortex activity might be greater in blindness because of the lack of competition from feedforward visual inputs and weakening of inhibitory mechanisms in the visual areas in this population^{41,42}. This view implies that our results were restricted to area V5/MT not because this area is especially sensitive to semantic properties of words, compared to other visual areas. Instead, the stronger results in this area were observed simply because we classified activity patterns for nouns and verbs, that is, two word classes with differing motion connotations. Other contrasts should reveal representations of other physical properties of word referents in other visual areas in blind individuals. Furthermore, based on this hypothesis, qualitatively similar, even if weaker results should be expected in sighted individuals. In line with this prediction, our analysis revealed statistical trends in area V5/MT in the sighted participants (Supplementary Fig. 4), which are qualitatively similar to the results observed in this area in the blind participants.

This view about activations induced in the blind visual cortex by language processing concurs with findings in the perceptual domain. Research in this domain shows that the processing of auditory and tactile stimuli by blind individuals activates specific visual areas that, in sighted individuals, process comparable stimuli in the visual modality⁴³⁻⁴⁵. Some of these auditory and tactile effects were also detected in the visual cortex of sighted individuals^{24,25,46-48}. Based on these results, it has been suggested that many visual areas can retrieve comparable information from perceptual experiences in different sensory modalities. Here, we show that this “computational equivalency” principle may also organize back projections from higher-level semantic regions to the visual cortex.

Furthermore, the above-described view is well aligned not only with our key findings, but also with several supplementary results we report. First, we found that the representation of differences between nouns and verbs in area V5/MT in blind individuals is not lateralized to the language-dominant hemisphere. Such lateralization can be expected based on the account that, in the absence of visual inputs, visual cortex is “colonized”⁴⁹ by signals from the language areas and, in essence, becomes a part of the language network. However, in the conjecture described above we argue that the original function of the back projections from the semantic system to the visual cortex is to support visual perception. In that case, the information carried by these back projections should reach visual areas in both hemispheres and the lateralization should not necessarily be expected. Second, the broad contrast between all concrete words (with physical referents) and all abstract words (without

physical referents) produced results in both dorsal and ventral visual areas in blind individuals, and even in the primary visual cortex (Supplementary Fig. 5). Our hypothesis predicts these areas are sensitive to specific physical properties of objects and actions conveyed by concrete words – those properties that are typically represented in these areas during feedforward visual processing.”

We now also strived to clarify our discussion of work by Seydell-Greewald and co-authors (lines 465-480):

“Conversely, in the visual cortex of sighted individuals, spoken words induced robust deactivation, relative to the rest periods. This effect might be driven by a known mechanism of inhibition of the visual system activity during auditory perception⁵⁴⁻⁵⁶. This process might be less efficient in blind individuals due to generally weaker inhibition mechanisms in their visual areas, described above. One study demonstrated that certain regions within the early visual cortex of sighted individuals are activated by spoken words⁵⁷. Here, we did not find such an effect. One explanation could be the differences in tasks used in the two studies. In the first experiment, Seydell-Greenwald and colleagues asked the participants to listen to sentences and decide whether they are semantically correct (e.g., “a big gray animal is an elephant”). In the two remaining studies, the authors asked the participants to listen to words and detect rare semantic oddballs (i.e., fruit names). In other words, in all experiments, the participants were encouraged to think about word meaning. In our study, we used a morphological transformation task, which focused the participants’ attention on word grammatical properties. Perhaps attention directed to word meaning can strengthen the modulation of the visual cortex by semantic regions, and result in above-rest activations for words also in the sighted visual cortex.”

I think the authors should be more straightforward in the abstract, intro, discussion etc... about what they mean by " These finding suggest that the blind visual cortex represents the physical properties of nouns and verbs....".

Please be explicit by what you mean by "physical properties ". Here authors test movement connotation. Does that mean that ANY other linguistic properties map onto ANY other occipital regions in blind? Surely this was not tested and such a statement is at odd with some literature so the phrasing could be more explicit.

As we now explicitly write in the discussion, our claim is that the processing of concrete words activates the semantic representations of word referents - objects and actions - in the high-level frontal and temporal regions. The information concerning the typical physical properties of these word referents are then back projected to the visual system. Our speculation is that the original function of this mechanism was to support visual perception by providing the visual system with information with predictive value. However, this mechanism might be preserved in blind individuals and, combined with increased sensitivity of the visual cortex in this population, drive strong responses to language in this region. Overall, our prediction is that responses to language in the blind visual cortex are not driven by bona-fide high-level linguistic representations; instead, they are driven by the representation of physical properties of word referents, which are retrieved from semantic representations computed in the high-level language regions.

Based on our results, we also make a topographic prediction. Specifically, we predict that various physical properties of objects and actions, retrieved from semantic representations, will be represented by specific visual regions in a way that respects the typical functional organization of the visual cortex for processing of visual inputs (and also other perceptual

inputs - tactile, auditory, etc.). Thus, motion connotations of objects and actions, conveyed by words, will be represented primarily in the dorsal areas, such as area V5/MT. Real-world size and animacy of the objects, named by words, will be represented in the ventral occipitotemporal cortex (Mahon et al., 2009; He et al., 2013). The shape of tools and other small objects, named by words, will be represented in the lateral occipitotemporal cortex (Peelen et al., 2013, 2014; Xu et al., 2023). What we think is the crucial aspect of our study is that it shows that these semantic effects are not accompanied by more abstract, grammatical or conceptual representations. This suggests that responses to language in the blind visual cortex can be indeed driven by relatively low-level, spatial and/or physical representations. We believe that this is an important result for researchers interested in the principles of plasticity in the human brain.

In summary, we suggest that the visual cortex receives physical properties of word referents rather than more abstract, conceptual properties of word referents or linguistic properties of words. We do not suggest that any visual area can represent any physical property of a word referent. Instead, we propose that specific visual areas in blind individuals remain sensitive to the same spatial properties that they typically process in sighted individuals. Thus, various physical properties of word referents can be represented across different visual areas. As we clarified in responses to the previous comment, we think that some of these physical properties can be projected back even to the early visual areas.

We now clarified our interpretation of the results in the abstract (note that we had to significantly shorten it, to comply with *Nature Communications* requirements):

"In blind individuals, language processing activates not only classic language networks, but also the "visual" cortex. What is represented in visual areas when blind individuals process language? Here, we show that area V5/MT in blind individuals, but not other visual areas, responds differently to spoken nouns and verbs. We further show that this effect is present for concrete nouns and verbs, but not abstract or pseudo nouns and verbs. This suggests that area V5/MT in blind individuals represents physical properties of noun and verb referents, salient in the concrete word category, but not conceptual or grammatical distinctions, present across categories. We propose that this motion-sensitive area captures systematically different motion connotations of objects (nouns) and actions (verbs). Overall, our findings suggests that responses to language in the blind visual cortex can be deconstructed to representing physical properties of words' referents, which are projected onto typical functional organization of this region."

We then elaborate on our interpretation of our results in the Discussion (lines 389-434), in paragraphs that we cited in responses to the previous comment.

Incidentally, we recently performed an independent fMRI study (currently under review) that further supports our theoretical perspective. We presented congenitally blind and sighted participants with 20 concrete words (nouns only). In an independent behavioral study, we collected ratings of physical and more abstract, conceptual similarity between the referents of these words. Using RSA, we showed representation of physical similarity between word referents throughout the visual cortex of blind individuals, including the early visual areas. Furthermore, we showed similar representation also in the visual cortex of sighted individuals. In contrast, more abstract, conceptual similarity was not represented in the visual cortex in either group. This is an independent confirmation that activations for words in the blind visual cortex might be driven by the representation of concrete, physical properties of word referents - the representation that is computed even in the sighted visual cortex.

Furthermore this study shows that, with a broader contrast (physical similarity, in general), the representation of physical properties of word referents can be detected throughout the visual cortex of blind individuals, including the early visual areas.

Reviewer #3 (Remarks to the Author):

This is a clearly written report of a nicely designed study of the processing of (nouns vs verbs) x (concrete vs abstract vs pseudo) in the brain of blind vs typical participants. The core claim of the authors is that MVPA in area V5 allows to decode nouns vs verbs, in the blind but not in the control group. My main concern is that this result appears statistically rather weak. Reading through the Results section, I noted the following points:

We would like to thank the reviewer for raising these important concerns. In the revised manuscript, we report new analyses, which further demonstrate the robustness of our key results.

We also clarify that our hypothesis makes a strong prediction about the type of properties that are represented in the blind visual cortex, following the presentation of spoken words. Specifically, we predicted that this region represents concrete, physical properties of word referents, but does not represent more abstract properties of word referents or grammatical properties of words themselves. We believe that our results support this prediction, even more so in the revised manuscript.

In contrast, based on our hypothesis, one should not expect strong differences in results between the blind and sighted individuals. As we strived to clarify in the revised manuscript, our hypothesis is that activations for linguistic stimuli in the blind visual cortex are driven by spatial representations of word referents which, in some form, are computed even in the sighted visual cortex. In the absence of visual inputs, these representations might be uncovered and potentially strengthened, which might result in activations for linguistic stimuli reported in this region in blind individuals. However, this hypothesis does not predict strong, qualitative differences in results obtained for words in the visual areas of both populations. In fact, in the revised manuscript we show that the removal of potential outliers from the data reveal statistical trends in the sighted group that are qualitatively similar to the results obtained in the blind group.

Overall, we agree that the between-group comparisons in our study are less robust than the between-condition comparisons. However, as we now clarify in the manuscript, this is actually in line with our hypothesis. Please see below for responses to specific comments.

- L 148: I appreciated that in the “omnibus analysis” Bonferroni correction was applied. Still I noted that contrary to what is stated in the methods not all occipitotemporal regions of the JuBrain atlas were apparently included (the LOC was omitted).

We apologize if this was unclear, but the lateral occipital area was included in all analyses, and is marked as “LO” (see Fig. 1 and Supplementary Fig. 2). The LO mask was created by combining the masks of areas hOc4la and hOc4lp (the two parts of the lateral occipital area: see Malikovic et al., 2016), provided by the JuBrain Anatomy Toolbox. No other mask of the lateral occipital region is provided by this toolbox.

We agree that, in the JuBrain Toolbox, the LO is defined quite posteriorly and dorsally (to avoid confusion, we now added a new supplementary figure to the manuscript, which presents the location of all visual ROIs used). We believe that this is how area LO is often defined in studies on visual perception. This is also how LO was defined in a classic paper

reporting activations for linguistic stimuli in the blind visual cortex (Bedny et al., 2011; the authors reported the following MNI coordinates for the left LO = -36 -90 -1). In contrast, researchers that study cross-modal and language-induced activations in the visual cortex often define the lateral occipital complex much more anteriorly, at the border of the occipital and the temporal lobe - here, we will name this more anterior region “the lateral occipitotemporal complex” (LOTCT).

We ran our analyses in the LOTCT, for completeness. There is no mask of the LOTCT in the JuBrain Anatomy Toolbox, likely because this region is usually defined by its functional rather than anatomical properties. Thus, we defined this region functionally, as a 10-mm sphere centered on a peak of preferential responses to words that refer to tools, relative to other semantic categories, in blind and sighted individuals, as reported by Peelen et al. (2013; Talairach coordinates: -50 -60 -5; these coordinates were transformed to MNI coordinates -53 -60 -12 using BiImage Suite - <https://bioimagesuiteweb.github.io/webapp/mni2tal.html>). However, we did not detect any significant effects in the LOTCT, in either the omnibus or the more detailed analysis (all corrected p values in the blind group > 0.25, all corrected values in the sighted group > 0.05).

We now describe these analyses in the new section “exploratory analyses”, in lines 317-329:

“We also further investigated whether differences between nouns and verbs were represented in high-level ventral and ventrolateral visual areas. To this aim, we ran our analyses in the lateral occipitotemporal cortex (LOTCT) and the ventral occipitotemporal cortex (VOTCT), as defined by previous studies that reported categorical effects in these regions in blind individuals^{36,37} (Supplementary Fig. 8). In the omnibus classification of activity patterns for all nouns and verbs, we found a significant effect in the VOTCT in the blind group (mean classification accuracy = 53,7%, $p = 0.024$). However, the more detailed analysis suggested that this effect is driven primarily by successful classification of pseudo nouns and verbs (mean classification accuracy = 55,9%, $p = 0.02$), in the absence of significant results for concrete and abstract nouns and verbs (both $p > 0.25$). We did not find robust effects in the LOTCT in either group (all p values > 0.1) or in the VOTCT in the sighted group (all p values > 0.25). Overall, the differences between concrete or abstract nouns and verbs do not seem to be robustly represented in ventral or ventrolateral visual areas.”

- L 152 I suspect that there is some “double dipping” in the (critical) comparison between groups of decoding in V5. Indeed, V5 was first selected by virtue of its high level of decodability in the blind, which then biases the subsequent comparison of decodability in the blind minus in controls.

In the omnibus analysis (Fig. 1), we indeed performed a between-group comparison only in area V5/MT. However, we did not choose this area because of the higher decoding accuracy in the blind group. As we now clarify in the Methods and the figure captions, we a priori decided to perform the between-group tests only in areas in which significant results were observed in either group. In this analysis, we observed significant results only in area V5/MT, thus the between-group comparison was performed only in this area. We do not think that this is a double dipping. We simply do not see the point of running between-group comparisons on the results that are not significant in either group. In such cases, we likely compare statistical noise with other statistical noise.

Incidentally, even assuming that we had bluntly performed the between-group comparisons in all visual ROIs in this analysis, the between-group comparison in area V5/MT would be strong enough ($p = 0.006$) to survive the Bonferroni correction across all ROIs (0.006×8 tests = 0.048). The effect in area V5/MT would be the only significant between-group effect across all visual ROIs. This is clear proof that the reported results are robust and that any form of double dipping did not affect them. We are happy to report this additional proof in the manuscript if the reviewer thinks that it can be useful for the readers.

- L 157: I guess that it is a just a consequence of the different decoding designs in the two cases, and I know that comparing p values is not the right way to go, but is it not surprising that decoding nouns vs verbs seems to be working less well when decoding was restricted to the supposedly decodable concrete items ($p=0.024$) than when it was applied to the whole set of data including abstract and pseudo items?

Actually, the accuracy of classification of only concrete nouns and verbs (55.9%) in area V5/MT in blind individuals was higher than the accuracy obtained in the omnibus analysis, which included nouns and verbs from all semantic categories (54.3%). It is true that this higher accuracy is accompanied by higher p value. This is likely the result of greater between-subject variability in results of classification of only concrete nouns and verbs, which is perhaps not surprising given that this analysis is performed on only 1/3 of our data.

Notably, we now report (lines 195-223; see also our response to the next comment) that, after the removal of potential outliers from the data, the accuracy of classification of activity patterns for concrete nouns and verbs rises to 57.2 % and the statistical significance of this contrast improves dramatically, too (testing against chance classification level, uncorrected $p = 0.001$, corrected $p = 0.003$)

Methodological considerations aside, in the manuscript, we suggest that area V5/MT is sensitive to motion connotations, which are systematically different across the referents of nouns and verbs. This is why this region primarily represents concrete nouns and verbs, for which this dimension is particularly salient. However, nouns and verbs from the abstract and the pseudo category can also differ in this dimension to some extent. In the case of abstract words, this is actually confirmed by the results of our behavioral experiment (see Table S3-4). Thus, perhaps nouns and verbs from other semantic categories also induced slightly different activity patterns in area V5/MT, which could have helped the classifier in the omnibus analysis. However, the more detailed analysis indicated that the results are significantly stronger - and detectable even with 1/3 of the collected data - for the concrete nouns and verbs. Such a pattern of results would be in line with our hypothesis.

To better visualize the interindividual variability in our data, we now present individual results in all critical analyses.

- L 163: consistent with my remark on line 152, this more stringent ANOVA shows no significant group difference in decodability in V5

As we now clarify in the manuscript, the critical contrast for our hypothesis is between the classification accuracies for nouns and verbs from different semantic categories (concrete, abstract, pseudo). The ANOVA shows the significant main effect of the semantic category - thus, we believe that we have strong statistical evidence to support the difference in results across the semantic categories.

As discussed above, we agree that the between group differences in the results in area V5/MT are not robust - we believe that this is actually in line with our hypothesis that the effects observed in blind individuals are driven by uncovering and strengthening of representations computed even in the sighted visual cortex, rather than by more dramatic, qualitative plasticity. In line with this hypothesis, we now show that removal of outliers for the data reveals statistical trends in the sighted group that are qualitatively similar to the results found in the blind group. We now describe this additional analysis in lines 195-223:

“Second, we reviewed the classification results obtained in area V5/MT for individual participants to ensure that our findings were not driven by outliers (Supplementary Fig. 4). In blind individuals, the above-chance classification of activations for concrete nouns and verbs in this area was observed in 14 out of 20 participants (70% of the participants). Across the results for three semantic categories in the blind group, 4 values were identified as potential outliers, defined as observations diverging from average classification accuracy in a given category by more than 2 standard deviations (1 value in the concrete category, 1 value in the abstract category, and 2 values in the pseudo category). Removing these values from the analysis further strengthened the group effects. Particularly, we still found above-chance classification of activations for concrete nouns and verbs (an increase in average classification accuracy from 55.9% to 57.2%, $p = 0.003$), but not for abstract and pseudo nouns and verbs (both p values > 0.25). In the direct comparison across the categories we again found that the classification accuracy obtained for concrete nouns and verbs was significantly higher than the average classification accuracy calculated across the abstract and pseudo nouns and verbs (mean difference = 7.6%, 95% CI [2.8%, 12.3%], $t(15) = 3.36$, $p = 0.004$, Cohen’s $d = 0.84$). In the sighted group, using the same data trimming procedure resulted in 2 values (1 value in the concrete category, 1 value in the abstract category) being removed from the data. Interestingly, after this procedure we found an uncorrected effect for concrete nouns and verbs also in area V5/MT in the sighted participants (an increase in average classification accuracy from 53% to 54.1%, uncorrected $p = 0.037$), in the absence of effects for the abstract and pseudo nouns and verbs (both uncorrected $p > 0.25$). The above-chance classification of activations for concrete nouns and verbs in this area was observed in 13 out of 20 sighted participants (13 out of 19 participants after data trimming, 68% of the participants). The direct comparison across semantic categories indicated a trend toward a higher classification accuracy for concrete nouns and verbs, compared to the average classification accuracy for abstract and pseudo nouns and verbs (mean difference = 6.3%, 95% CI [-0.2%, 12.7%], $t(18) = 2.05$, $p = 0.056$, Cohen’s $d = 0.48$). Overall, the analysis of trimmed data suggests that, while statistically weaker, the results in area V5/MT in the sighted group might be qualitatively similar to those found in the blind group.”

We agree that the differences in results (or lack thereof) across groups, and their implications for our hypothesis, was not sufficiently discussed. We now better explain our stand on this issue in the revised manuscript. Throughout the Discussion, we also strived to clarify our interpretation of the critical results itself.

In lines 341-434, we write:

“Words are highly multidimensional objects. A given brain area can represent differences between words because of how they sound (phonological dimension), what they mean (semantic dimension), or what role they play in a sentence (grammatical dimension). The aim of our study was to investigate which word properties are represented in the visual cortex of blind individuals, and could drive activation for linguistic stimuli in this region⁵⁻¹⁰. We

focused on a fundamental linguistic distinction, that between nouns and verbs, and investigated whether semantic or grammatical aspects of this distinction are represented in the blind visual cortex. We found above-chance classification of activity patterns for nouns and verbs in visual area V5/MT in the blind participants, but not in other visual areas in this group. We further showed that the effect in area V5/MT in the blind was primarily driven by successful classification of activations for concrete nouns and verbs, in the absence of significant results for abstract and pseudo nouns and verbs. Different classification results for nouns and verbs from different semantic categories cannot be explained by differences in auditory representations, as the phonological properties of words were not systematically different across the semantic categories. Similarly, this pattern of results cannot be driven by grammatical representations – in all semantic categories, words were readily recognizable as nouns or verbs. Thus, these results suggest that area V5/MT in blind individuals represents differences between nouns and verbs because of their differing semantic properties, that is, through representations of objects and actions named by words.

Our study further shows that area V5/MT in blind individuals captures properties that are saliently and systematically different for concrete noun and verb referents, but not necessarily for abstract noun and verb referents. We suggest that the most plausible candidate for such a property is the representation of motion connotations, which is likely differently activated by concrete nouns, generally naming stationary objects, and concrete verbs, generally naming dynamic actions. Area V5/MT is sensitive to visual²³ and auditory^{24,25} motion, with auditory sensitivity being preserved and elevated in blind individuals²⁶⁻²⁸. Our study suggests that this area can also retrieve motion connotations of objects and actions from semantic representations. In other words, our findings indicate that this area can retrieve the same physical property from memory-driven semantic representations and stimulus-driven perceptual (visual, auditory, etc.) representations.

This conclusion is in line with previous reports that, in both blind and sighted individuals, high-level ventral visual regions respond differently to words referring to objects of different shape and size³⁸. Similar effects were shown in these regions during the presentation of object pictures³⁹. Here, we show that also dorsal visual regions can use information conveyed by spoken words to perform their relatively typical computations, such as representation of motion and motion connotations. Crucially, we used abstract and pseudo words to test for more abstract, conceptual or grammatical representations in the blind visual cortex, as such representations could be potentially computed on top of simpler representations of physical properties. However, we did not find any clear sign of such abstract representations in any visual area tested.

Overall, our results suggest that, during language processing, the blind visual cortex represents the physical properties of word referents, more salient in the concrete word category, rather than more abstract linguistic features, present across the word categories. The topography of effects observed in our study – that is, finding the representation of differences between two word classes with systematically different motion connotations primarily in the motion-sensitive area V5/MT - indicates that these physical connotations conveyed by words are mapped onto the typical functional organization of the visual cortex, present also in the sighted brain.

Our findings suggest that, after the activation of semantic representations in higher-level brain regions, the physical properties of word referents, retrieved from these representations, are back projected to the visual system in a way that parallels feedforward visual processing.

Thus, the “motion template” of objects and actions is back projected to the dorsal stream areas, such as area V5/MT, the “shape template” is back projected to the ventral stream areas, and so on. Such organization of the back projections from the semantic system might be most useful for forming visual predictions and, consequently, supporting visual perception⁴⁰. This mechanism might be preserved and functional in blind individuals, even if its original function is not relevant in this population. Moreover, the impact of such modulations on the visual cortex activity might be greater in blindness because of the lack of competition from feedforward visual inputs and weakening of inhibitory mechanisms in the visual areas in this population^{41,42}. This view implies that our results were restricted to area V5/MT not because this area is especially sensitive to semantic properties of words, compared to other visual areas. Instead, the stronger results in this area were observed simply because we classified activity patterns for nouns and verbs, that is, two word classes with differing motion connotations. Other contrasts should reveal representations of other physical properties of word referents in other visual areas in blind individuals. Furthermore, based on this hypothesis, qualitatively similar, even if weaker results should be expected in sighted individuals. In line with this prediction, our analysis revealed statistical trends in area V5/MT in the sighted participants (Supplementary Fig. 4), which are qualitatively similar to the results observed in this area in the blind participants.

This view about activations induced in the blind visual cortex by language processing concurs with findings in the perceptual domain. Research in this domain shows that the processing of auditory and tactile stimuli by blind individuals activates specific visual areas that, in sighted individuals, process comparable stimuli in the visual modality⁴³⁻⁴⁵. Some of these auditory and tactile effects were also detected in the visual cortex of sighted individuals^{24,25,46-48}. Based on these results, it has been suggested that many visual areas can retrieve comparable information from perceptual experiences in different sensory modalities. Here, we show that this “computational equivalency” principle may also organize back projections from higher-level semantic regions to the visual cortex.

Furthermore, the above-described view is well aligned not only with our key findings, but also with several supplementary results we report. First, we found that the representation of differences between nouns and verbs in area V5/MT in blind individuals is not lateralized to the language-dominant hemisphere. Such lateralization can be expected based on the account that, in the absence of visual inputs, visual cortex is “colonized”⁴⁹ by signals from the language areas and, in essence, becomes a part of the language network. However, in the conjecture described above we argue that the original function of the back projections from the semantic system to the visual cortex is to support visual perception. In that case, the information carried by these back projections should reach visual areas in both hemispheres and the lateralization should not necessarily be expected. Second, the broad contrast between all concrete words (with physical referents) and all abstract words (without physical referents) produced results in both dorsal and ventral visual areas in blind individuals, and even in the primary visual cortex (Supplementary Fig. 5). Our hypothesis predicts these areas are sensitive to specific physical properties of objects and actions conveyed by concrete words – those properties that are typically represented in these areas during feedforward visual processing.”

- L 179: I regretted that the separate results of the R and L hemispheres were never reported, particularly for V5 of course. In the field of language, pooling a priori the L and R ROIs without even looking into their differences is a bit unusual.

We now investigate the lateralization of the results. However, as we discussed in the responses to Reviewer 2, we believe that simply investigating the left and the right hemisphere might be misleading in our study, since the language network is known to be less lateralized in blind individuals than in sighted individuals. This effect is primarily driven by a greater interindividual variability in the blind group - a significant portion of blind individuals actually show comparable activations for linguistic stimuli in both hemispheres, or even stronger activations in the right hemisphere (Lane et al., 2015; Lane et al., 2017; see also Röder et al., 2000, 2002; Dzięgiel-Fivet and Jednoróg, 2024). Thus, testing the hypothesis that the effects in our study are stronger in the “language-dominant” hemisphere first requires empirically determining what hemisphere it is in each participant - especially in each blind participant. Using the left hemisphere as “language-dominant” could induce a bias in comparisons between sighted and blind individuals, as that would capture the language lateralization better in the sighted group than in the blind group.

We tested if these lateralization differences can be detected also in our data. For each participant, we calculated a simple “lateralization index” of activations for language. We first averaged the activations for all words and pseudowords used in the study across three classic language regions: the left superior temporal cortex (Area TE 3 in the JuBrain Anatomy Toolbox), the left BA 44, and the left BA 45. We then subtracted the obtained value from the average activation calculated across the analogous regions in the right hemisphere. Thus, for each participant, the value greater than 0 indicated left-lateralization of the activations for language in the language network, whereas the value lower than 0 indicated right-lateralization.

Indeed, our lateralization index indicated significant left-lateralization of activations for language in the sighted group (mean lateralization index values = 0.157, $t = 4.89$, $p < 0.001$). In contrast, no significant lateralization was detected in the blind group (mean lateralization index values = 0.059, $t = 1.54$, $p = 0.14$). A direct comparison confirmed greater lateralization of activations for language in the sighted participants, compared to the blind participants (trend level, $t = 1.97$, $p = 0.056$). The lateralization indices showed that, in 6 out of 20 blind participants, linguistic stimuli activated the right analogs of classic language regions more strongly. In sighted participants, only 2 out of 20 participants showed this pattern of results. This is in line with reports that, in the typical population, language is left-lateralized in approximately 92 % of individuals (results from right-handers only: Knecht et al., 2000; results from both right-handers and left-handers: Labache et al., 2023), and shows that our basic lateralization measure is actually quite accurate.

We used individual language lateralization indices to run, in each participant, the analysis in the language dominant hemisphere and in the language non-dominant hemisphere separately. However, the results obtained in this analysis were weaker than those produced by the bilateral analysis. We now report these results in lines 298-316:

“We tested whether the effects observed in the classification analysis in area V5/MT were lateralized to the language-dominant hemisphere. Given that the lateralization of the language network is more variable in blind individuals than in sighted individuals^{6,9,33-35}, we empirically determined which hemisphere is language-dominant in each participant by comparing the magnitudes of activations for words and pseudowords in classic language regions (superior temporal and inferior frontal cortices) and in their analogs in the right hemisphere (see Methods). For each participant, we then ran separate classification analysis in area V5/MT in the language-dominant and the language-nondominant

hemisphere. However, the group effects obtained in these analyses (Supplementary Fig. 7) were weaker than those produced by the bilateral analysis. In the language-dominant hemisphere, we observed uncorrected effects in the omnibus classification of activity patterns for all nouns and all verbs in both blind individuals (mean classification accuracy = 52.3%, uncorrected $p = 0.038$) and sighted individuals (mean classification accuracy = 52%, uncorrected $p = 0.049$). However, these results did not survive the correction for multiple comparisons (both p values > 0.15). Furthermore, no significant results, in either group and hemisphere, were detected in the more detailed analysis, in which we classified activity patterns for nouns and verbs from each semantic category separately (all p values > 0.1). This suggests that the robust results reported in area V5/MT in blind individuals in the main analysis were driven by activity patterns in both hemispheres.”

As we now discuss, this analysis further challenges the account that, in the absence of visual inputs, the visual cortex is “colonized” (Bedny, 2017) by signals from the language areas and, in essence, becomes a part of the language network. In that case, one could expect the results to be stronger in the language-dominant hemisphere. In the manuscript, we develop a different theoretical perspective - we propose that, in sighted individuals, the physical properties of word referents are back projected to the visual system in order to prepare the visual system for the likely incoming stimulation (i.e., to predict incoming visual information; Rao and Ballard, 1999). When you hear “watch out for the ball” it is likely that the ball will enter your visual field in a moment, and it might be adaptive to prepare the visual system for a quick detection of this object. We propose that this mechanism is preserved in blind individuals and, combined with weakening of inhibitory mechanisms in the blind visual cortex, might drive strong responses to language in this region.

Based on our theoretical perspective, it is only expected that differences between concrete nouns and verbs in area V5/MT of blind individuals are comparably represented in the language-dominant and the language non-dominant hemisphere, and that, consequently, the results are stronger in the bilateral analysis. If the original goal of this mechanism was to support vision, then this information should be projected to both hemispheres

We now explain these important points in the Discussion. We first clarify our theoretical view on the results in the paragraphs cited in the responses to the previous comment. Then, in lines 419-434, we write:

“Furthermore, the above-described view is well aligned not only with our key findings, but also with several supplementary results we report. First, we found that the representation of differences between nouns and verbs in area V5/MT in blind individuals is not lateralized to the language-dominant hemisphere. Such lateralization can be expected based on the account that, in the absence of visual inputs, visual cortex is “colonized”⁴⁹ by signals from the language areas and, in essence, becomes a part of the language network. However, in the conjecture described above we argue that the original function of the back projections from the semantic system to the visual cortex is to support visual perception. In that case, the information carried by these back projections should reach visual areas in both hemispheres and the lateralization should not necessarily be expected. Second, the broad contrast between all concrete words (with physical referents) and all abstract words (without physical referents) produced results in both dorsal and ventral visual areas in blind individuals, and even in the primary visual cortex (Supplementary Fig. 5). Our hypothesis predicts these areas are sensitive to specific physical properties of objects and actions

conveyed by concrete words – those properties that are typically represented in these areas during feedforward visual processing.”

- L 180: one may regret that V5 was not identified individually with some movement localizer, as the location of V5 varies notably across individuals, and maybe more so between blind and typical individuals (doi: 10.1093/cercor/bhw180).

We agree that, in general, individual localization of regions of interest is desirable. At the same time, we believe that such an approach would not be optimal for our research question. Specifically, individual localization of area V5/MT, but not other visual areas, would not give “a fair chance” to the other visual areas in our analyses. Such a bias would make any claims concerning topographic specificity of our effects doubtful.

Nevertheless, even using less precise, atlas-based localization, we still found the effect of interest in area V5/MT. At the level of individual results, the above-chance classification of activations for all nouns and all verbs in this area was observed in 16 out of 20 blind participants in the omnibus analysis (80 % of the blind participants). The above-chance classification of activations for concrete nouns and verbs in the analysis in specific semantic categories was observed in 14 out of 20 blind participants (70 % of the blind participants). We can only expect that these results would be even stronger with a more precise, individual localization approach. We now present these individual data in the figures.

One potential concern related to the atlas-based localization of area V5/MT is that the decoding in this area might be driven by a relatively small number of voxels, which in fact belong to different functional areas, and would be excluded with a more precise, individual localization approach. We now present a new analysis which shows that this is not the case in our study. We randomly drew (without replacements) 40, 80, 120 and 160 voxels from our bilateral V5/MT mask and ran the classification analyses only in these voxels. We performed 1000 iterations of this analysis and we averaged the results for each number of voxels. We found above-chance classification of activations for concrete nouns and verbs in blind individuals across all analysis levels (corrected p value for the analysis on 40 voxels = 0.068; corrected p values for all other analysis levels < 0.05). Thus, this effect does not depend on several specific voxels within the V5/MT mask, and can be reliably detected across a broad spectrum of analysis parameters. In contrast, no significant effects, at any analysis level, were observed for abstract and pseudo nouns and verbs (all p values > 0.25).

We believe that this new analysis shows that our results in area V5/MT of blind individuals are robust and cannot be driven by imperfections in V5 localization. We now describe it in lines 180-194:

“We investigated the robustness of our results in area V5/MT with two additional analyses. First, we wanted to rule out that the observed results are driven by only a small subset of voxels within the V5/MT mask – for example, only those voxels that border more anterior areas that represent actions at increasingly conceptual level³². To investigate this possibility, we iteratively drew subsets of voxels (40, 80, 120, and 160 voxels, 1000 random draws at each level) from the V5/MT mask and performed the classification of activations for nouns and verbs from each semantic category in only these subsets (Supplementary Fig. 3). We found above-chance classification of activations for concrete nouns and verbs in blind individuals across all analysis levels (p value for the analysis on 40 voxels = 0.068; all other p values < 0.05). Thus, this effect does not depend on several specific voxels within the V5/MT mask, and can be reliably detected across a broad spectrum of analysis parameters.

In contrast, no significant effects, at any analysis level, were observed for abstract and pseudo nouns and verbs (all p values > 0.25). As in the main analysis, no significant effects for nouns and verbs from any semantic category were observed in the sighted group (all p values > 0.25)."

- L 189: the searchlight analysis did not confirm the results of the ROI analysis in V5

We now discuss in the manuscript why the searchlight analysis did not detect significant effects in area V5/MT, which were detected in the ROI analysis. This difference in results across these two methods is likely driven by the fact that searchlight analysis included voxels from only one hemisphere at a time. At the same time, our analysis of laterality, performed at the reviewer's request, shows that only the analysis performed on activity patterns from both hemispheres results in robust effects. The robustness of results from both hemispheres is now confirmed with several additional analyses, described in responses to previous comments..

We now explain this issue in the manuscript, and acknowledge that the disparity between the results of the ROI and the searchlight analysis can be seen as one of limitations of our study (lines 499-508):

"Two limitations of our work should be acknowledged. First, the searchlight analysis did not confirm the results obtained in area V5/MT in the analysis in visual regions of interest. This difference in results across these two methods is likely driven by the fact that the searchlight analysis included V5/MT voxels from only one hemisphere at a time. Our analyses show that only including activity patterns from both hemispheres results in robust effects. Second, we did not confirm that area V5/MT represents motion connotations of word referents with representational similarity analysis. Our study was optimized for multi-voxel pattern classification and used a block design and different words in each experimental run. While these methodological choices served an important purpose in our study, they also precluded the representational similarity analysis."

- L 200-206: the authors report nice results, without however further supporting the existence of a significant group difference in V5.

Please see below for our joint response to this and the next comment.

- L 217: the decoding of concrete vs abstract words, which was predicted to "accompany" the decoding of concrete verbs vs nouns, was significant in V4 and V5, but with a group difference restricted to V4, and not V5.

As we explained in responses to the previous comments, we now discuss in the manuscript the issue of differences in results (or lack thereof) across groups. We agree that this difference is not robust, but our hypothesis does not expect it to be robust. As we now clarify in the manuscript, our hypothesis is that effects for linguistic stimuli observed in the blind visual cortex are driven by uncovering and strengthening of representations that are present even in the sighted visual cortex. This is suggested by previous studies showing similar semantic effects induced by spoken words in the ventral areas of blind and sighted individuals (He et al., 2013; Peelen et al., 2013, 2014). As we described above, this is also to some extent suggested by our analysis of trimmed data. described in lines 195-223, and cited in responses to previous comments. This analysis showed statistical trends in the sighted group that are qualitatively similar to the results observed in the blind group.

We now clarify our interpretation of the results, and the issue of between-group differences, in lines 389-434, cited in our responses to the previous comments.

Reviewer #2 (Remarks to the Author):

The authors should be praised for their thorough and thoughtful reply.

I find the clarification that the noun-verb decoding is selective to the concrete grammatical class and selective only to the V5/MT ROI convincing.

I only have a few remaining points of clarification.

We would like to thank the Reviewer for this positive assessment of our revised manuscript. We provide our responses to remaining comments below.

One thing that remains unclear is that the region of interest analyses were performed using maps from the JuBrain Anatomy Toolbox (Eickhoff et al., 2005). All ROIs were defined bilaterally. However all those regions have different sizes and therefore a different number of voxels as features for the decoder. How does that impact the results?

Our visual ROIs indeed had different sizes, with the V5/MT ROI being the smallest (see Supplementary Table 5, copied below). While this is a natural consequence of using anatomical masks for ROI definitions, we agree that this raises methodological question described by the Reviewer.

Supplementary Table 5. Number of voxels included in each visual ROI. LO - the lateral occipital area; FG - the fusiform gyrus.

V1	V2	V3	V4	V5	V6	LO	FG
2441	2031	2680	2070	193	262	2366	2498

To investigate this issue, we drew the same number of voxels (i.e., 193 voxels – the size of the V5/MT mask) from each visual ROI and reran our key analyses in these subsets. We performed 1000 iterations of this analysis for each participant and averaged the results across the iterations. We replicated the results reported in the manuscript (see Supplementary Figures 3 and 4). In the initial analysis, performed with words from all semantic categories combined, the classification of words into nouns and verbs was still above chance level only in area V5/MT in the blind participants. In the more detailed analysis, performed within specific semantic categories, the classification of concrete nouns and verbs was still above chance level only in area V5/MT in blind participants (with qualitatively similar trends detected in sighted participants after removal of outliers - see Supplementary Figure 6). Moreover, the classification of abstract nouns and verbs was still not successful in any visual ROI in either group.

As a sanity check, we applied the above-described procedure to the superior temporal cortex ROI (which originally contained 991 voxels). We still obtained successful classification of concrete, abstract and pseudo nouns and verbs in this region, in both groups (see Supplementary Figure 7). Notably, also the analysis performed in subsets of voxels from the V5/MT mask (see Supplementary Figure 5), which we prepared when revising the manuscript for the first time, confirms the viability of this analytical approach.

In summary, this new analysis suggests that the results we report are not driven by the differences in the sizes of anatomical masks that were used. We now described this new control analysis in lines 180-188 of the manuscript:

We investigated the robustness of our results in area V5/MT with several additional analyses. First, we asked whether the observed topographic specificity of the effect found in this area could be explained by the fact that its anatomical mask is smaller than the masks for other visual areas (see Supplementary Table 5). To investigate this issue, we iteratively drew the same number of voxels (193 voxels – the size of the V5 mask) from the mask of each visual region and reran our key analyses in these subsets (Supplementary Figures 3 and 4). We replicated the results reported in the main analysis. This suggests that the reported results are not driven by the differences in the sizes of anatomical masks that were used.

In an additional analysis, classification of activity patterns for nouns and verbs from specific semantic categories was performed in subsets of voxels from the V5/MT mask. These subsets of voxels (40, 80, 120, 160 voxels) were iteratively drawn from the V5/MT mask. Were those different voxel sets randomly chosen from the bilateral ROIs? This is surprising this works even with 40 voxels if taken from a bilateral mask, while it does not work with a unilateral mask including much more voxels.

Yes, the voxels in this analysis were randomly drawn from bilateral ROIs. As described above, the bilateral V5 ROI consisted of 193 voxels, with the left-hemisphere V5 ROI including 86 voxels (~45% of all voxels) and the right-hemisphere V5 ROI including 107 voxels (~55% of all voxels). In this context, two points are worth considering

(1) If, for simplicity, we assume that the draw probabilities remain unchanged during drawing (see below for elaboration on this issue), then whether a given voxel is drawn from the left or the right hemisphere essentially becomes a coin flip with slightly skewed probability (55% vs. 45%). With these parameters, even when we draw only 40 voxels, the probability of drawing all voxels from the same hemisphere is less than 1/24 000 000 000. The probability of drawing at least 30 (75%) voxels from the same hemisphere is ~1/135 (we obtained these values in: <https://www.omnicalculator.com/statistics/coin-flip-probability>). These probabilities are even lower in our actual analysis because we drew without replacements – in this procedure the probability of drawing from the same hemisphere becomes lower with each “successful draw” because there are less voxels remaining in this hemisphere. In summary, almost all iterations of our analysis were likely to contain a significant representation of voxels from both hemispheres. Thus, while we agree that the theoretical explanation of why the decoding in our study works well only when the voxels from both hemispheres are included might not be straightforward (see our response to the next comment), we are not surprised that the analysis in question produced significant results even when the analyses in separate hemispheres did not.

(2) It is true that the decoding in the bilateral analysis worked even with 40 voxels, whereas the decoding in separate hemispheres did not produce significant results with 86 or 107 voxels. However, the bilateral analysis actually suggests that such increase in number of voxels produces only modest increase in statistical power – the classification of concrete nouns and verbs in bilateral masks in the blind participants reached $p = 0.022$ in the analysis with 40 voxels and $p = 0.015$ in the analyses with 80 and 120 voxels.

Related to that point, the fact that representations of concrete verb vs sounds rely on bilateral networks is intriguing. I am however not fully convinced by the argument provided by the authors. The fact that the process recruits both hemispheres is different from the fact that decoding only works if patterns of activity include both hemispheres. For instance, motion direction is encoded in left and right V5/MT but directions can be decoded in each of them

separately. Why would the process necessarily engage a representation that requires both hemispheres to be reliably decoded?

We would like to clarify that it has never been our intention to suggest that successful decoding of concrete nouns and verbs in area V5/MT always requires activity patterns from both hemispheres – while we used a potentially misleading term “ground truth” in our previous responses (we now ensured that we do not use it in the manuscript), we only meant “ground truth of our study”, which has certain statistical power to detect effects of interest. We do expect that other studies will show that this effect can be detected even based on activity patterns from only one hemisphere. We now clarify this in lines 323-325 of the manuscript:

One might speculate that the analysis in specific hemispheres lacked power to detect robust effects, akin to those detected in the bilateral analysis.

Furthermore, we removed from the discussion the suggestion that our analysis in specific hemispheres might indicate that the V5/MT representation of differences between concrete nouns and verbs is not lateralized. If we truly lack power to detect robust effects in specific hemispheres, then perhaps this particular analysis should not be used to make theoretical claims.

Authors suggest that responses to language in the blind visual cortex can be driven by relatively low-level, spatial and/or physical representations. This suggests that the visual cortex receives physical properties of word referents rather than more abstract, conceptual properties of word referents or linguistic properties of words. I sympathize with this idea. But how do the authors privilege this idea rather than the possibility that occipital regions implement semantic representations stored as abstract knowledge independent of simulating the physical features of what a specific word refers to? I understand it is selective for motion but could it be an "abstract" representation of implied motion rather than the simulation of physical motion in the visual (sighted) or tactile, auditory (blind) modalities?

Thank you for this interesting comment. Perhaps the best way to disentangle these two hypotheses is to study neural representation of color – a physical feature that, in the case of profound and congenitally blindness, cannot be “sensory simulated”. A study on this issue (Wang et al, 2020, *Neuron*) presented sighted and congenitally blind individuals with words that name various fruits. In sighted individuals, the representation of fruit color was found both in the visual areas and in the anterior temporal lobe. In contrast, in congenitally blind individuals, representation of fruit color was found only in the anterior temporal lobe. This suggests that actual sensory experience with (or ability to sensory simulate) a given physical feature is necessary for its representation to emerge in the visual cortex, also in blind individuals. Abstract knowledge that “apples are red, and similar in color to strawberries”, which most blind individuals possess (see Wang et al, 2020), is not represented in this region.

We do not see reasons why motion connotations, retrieved from semantic representations following the word presentation, should be represented in the blind visual cortex following different principles. In any case, our main finding is that the visual cortex in blind individuals represents physical features of word referents, but not conceptual or grammatical distinctions conveyed by spoken words. Based on this finding, we write that “our findings suggest that responses to language in the blind visual cortex can be deconstructed to representing physical properties of words’ referents, which are projected onto typical functional

organization of this region.” While we agree that the question raised by the Reviewer is important, we also believe that these findings remain valid irrespective of the precise format in which the physical features of word referents reach the visual cortex.

We now comment on this interesting issue in lines 380-395 of the manuscript:

This conclusion is in line with previous reports that, in both blind and sighted individuals, high-level ventral visual regions respond differently to words referring to objects of different shape and size³⁷. Similar effects were shown in these regions during the presentation of object pictures to sighted participants³⁸. Importantly, representation of fruit color, following the spoken presentation of fruit names, was documented in the ventral visual cortex of sighted individuals, but not congenitally blind individuals³⁹. This result suggests that visual areas only represent object physical features that are grounded in individual perceptual experience, also in blind individuals – abstract semantic knowledge that “apples are red, and similar in color to strawberries” is not represented in the visual cortex in this population. These findings concur with the results of our study. First, we show that also dorsal visual regions can use information conveyed by spoken words to perform their relatively typical computations, such as representation of motion and motion connotations. Second, we used abstract and pseudo words to test for more abstract, conceptual or grammatical representations in the blind visual cortex, as such representations could be potentially computed on top of simpler representations of physical properties. However, we did not find any clear sign of such abstract representations in any visual area tested.

Given that classification is significant only between concrete nouns and verbs; what is the meaning of Fig. 1 which, if I am not mistaken, mix the different grammatical classes ?

The aim of this initial analysis was to investigate, with maximal statistical power (i.e., with all semantic categories combined), what visual regions might represent the differences between nouns and verbs. Even in this analysis we obtained significant results only in area V5/MT in the blind participants. Subsequently, in a more specific analysis, we showed that this effect is primarily driven by concrete nouns and verbs. We would prefer to keep both the initial and the specific analysis in the manuscript as this reflects how we planned the data analysis and actually analyzed the data.

As we explained in our responses to the previous round of comments, we do not think that finding an effect in area V5/MT in the initial analysis is at odds with our hypothesis. In the manuscript, we suggest that area V5/MT is sensitive to motion connotations, which are systematically different across the referents of nouns and verbs. This is why this region primarily represents differences between concrete nouns and verbs, for which this dimension is particularly salient. However, nouns and verbs from the abstract and the pseudo category can also differ in this dimension to some extent. In the case of abstract words, this is actually confirmed by the results of our behavioral experiment (see Supplementary Tables 3 and 4). Thus, perhaps nouns and verbs from other semantic categories also induced slightly different activity patterns in area V5/MT, which could have helped the classifier in the initial analysis. However, the more detailed analysis indicated that the results are significantly stronger - and detectable even with 1/3 of the collected data - for the concrete nouns and verbs. Such a pattern of results would be in line with our hypothesis.

Reviewer #3 (Remarks to the Author)

The authors made commendable efforts to address the points I raised in the first round of reviews. They confirm my observation that the differences between blind and seeing participants are not entirely water tight, which, although the theoretical discussion accommodates this state of affairs, reduces somewhat the originality of the work. Regarding my observation on “double dipping”, the authors provide two answers. I believe that the first one is not quite correct, in as much as they DO select V5 based on significant decoding in the blind. It would not be double dipping if it had been selected based on decoding accuracy in the average of both groups. They provide a second, better response, based on the fact that the group difference was significant in V5 even when Bonferroni-corrected, something which they should put in the paper.

We would like to thank the Reviewer for the positive assessment of our revisions. Following this remaining comment, we described the requested, additional sanity check on statistical quality of our results in the manuscript. In lines 688-693 we now write:

The between-group tests were performed only in areas in which significant results were observed in either group. Since we observed significant results only in area V5/MT in the blind participants, the correction for multiple comparisons was not necessary. Notably, the between-group difference reported in area V5/MT remained statistically significant even after Bonferroni correction across the visual areas (i.e., correction for 8 tests).

We beg to differ in our assessment of the finding that area V5/MT might compute qualitatively similar representations of noun and verb referents in congenitally blind and sighted individuals. So far, comparable representations of word referents have been primarily documented in the ventral visual cortices of these two populations. Here, we show that such effects can be found also in other visual regions (see lines 379-394 of the manuscript for further discussion).

We believe that this finding further supports the claim that responses to language in the visual cortex, found in blind individuals, might be driven by relatively typical visuospatial representations computed in this region.